# DeepHA: Scaling Action Chains Elicits Deep Hierarchical Agents

Zihao Wang [* 1]  Muyao Li [* 1]  Kaichen He [1]  Haowei Lin [1]  Xiaojian Ma [2]  Anji Liu [3]  Yitao Liang [1]

## Abstract

Existing autonomous agents are often constrained by a single, predefined action space, which limits their generalization capabilities across diverse tasks and can introduce compounding errors through decoupled policy execution. To address these limitations, we introduce the Deep Hierarchical Agent (DeepHA), a unified architecture that operates across a mixture of heterogeneous action spaces, flexibly generating actions ranging from high-level semantic skills to low-level motor controls. We further propose an action hierarchy reasoning framework, which enables the agent to use higher-level abstract actions as structured 'thoughts' to guide the generation of low-leval actions. To manage the computational demands of this deep reasoning in long-horizon tasks, we develop a memory-efficient mechanism that dynamically compresses historical context and leverages Key-Value (KV) caching, reducing context length by approximately 75% without sacrificing performance. We conduct extensive evaluations on a new, large-scale benchmark of over 800 diverse Minecraft tasks. Results show that DHA significantly outperforms prior methods, establishing a new state-of-the-art and demonstrating superior generalization, particularly in complex, multi-step planning tasks.

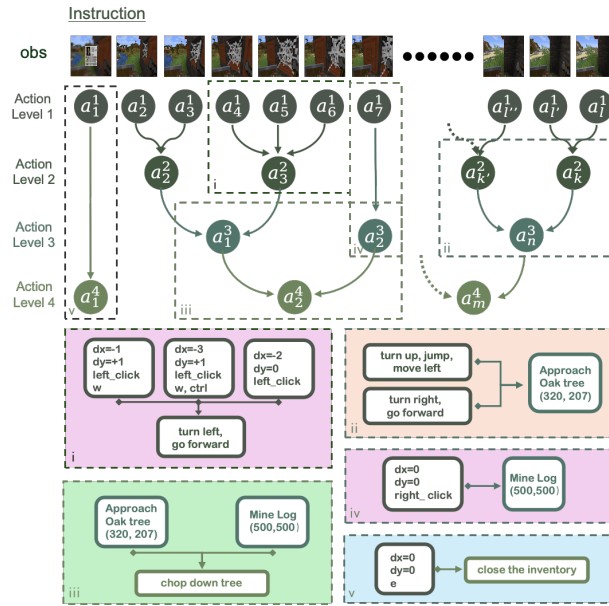

*Figure 1.* An illustration of the multi-level action hierarchy for the instruction "BUILD A HOUSE" in Minecraft. A sequence of visual observations ($o_t$) is temporally aligned with a four-tiered action space. This hierarchy ranges from **Level 4** high-level semantic skills (e.g., `Mine Log`) and **Level 3** grounding actions that specify object targets (e.g., `Approach Oak tree (320,207)`), down to **Level 2** mid-level motion policies (e.g., `turn left, go forward`), and finally to **Level 1** primitive keyboard and mouse operations (e.g., `mouseMove(-1,2)`, `press(w)`). The figure exemplifies how a single, complex goal is decomposed into progressively finer-grained sub-actions, highlighting the temporal and semantic abstraction inherent in the action space.

## 1. Introduction

Prevailing architectures for autonomous agents often adopt a hierarchical design where a high-level Large Language Model (LLM) generates an abstract action from a manually

---
[*]Equal contribution  [1]Institute of Artificial Intelligence, Peking University, Beijing, China [2]Beijing Institute of Artificial General Intelligence, Beijing, China [3]School of Computing, National University of Singapore, Singapore. Correspondence to: Zihao Wang <zhwang@stu.pku.edu.cn>, Muyao Li <2350076251@qq.com>, Anji Liu <anjiliu@nus.edu.sg>, Yitao Liang <yitaol@pku.edu.cn>.

*Proceedings of the 43$^{rd}$ International Conference on Machine Learning*, Seoul, South Korea. PMLR 306, 2026. Copyright 2026 by the author(s).

predefined action space (Driess et al., 2023; Belkhale et al., 2024b; Gu et al., 2023; Liang et al., 2022; Brohan et al., 2022b). This abstract action is then translated into low-level, executable environment actions by a separate, independently trained policy or a fixed API decoder (Qin et al., 2025; Chen et al., 2021; Wang et al., 2024a; 2025b). However, this paradigm faces two critical limitations. First, it relies on a single, monolithic action space (e.g., language skills, motion primitives), yet research has shown that the optimal choice of action abstraction is highly task-dependent; an action space that excels at navigation may be ill-suited for precise object manipulation. Consequently, no single action space can enable an agent to achieve state-of-the-art performance across a diverse range of benchmarks (Wang et al.,

2025a; Team, 2025b). Second, the decoupled nature of the high-level LLM and the low-level policy can introduce compounding errors, where inaccuracies in the generated abstract action are magnified by an imperfect decoder (Wang et al., 2024c; Driess et al., 2023; Brohan et al., 2022b).

To overcome these constraints, we propose a paradigm shift from an agent reliant on a single, predefined action space to one that can dynamically reason and operate across multiple action spaces during inference. We introduce the **Deep Hierarchical Agent** (DeepHA), which features a high-level VLM capable of generating actions from a *mixture of heterogeneous action spaces*—spanning high-level language skills, coordinate-based grounding actions, mid-level motion commands, and low-level raw action sequences. These are interpreted by corresponding specialized policies within a Mixture-of-Policies (MoP) framework, allowing DeepHA to autonomously select and execute the most appropriate action abstraction for any given task or sub-task. This design effectively unifies the strengths of direct VLM models and structured hierarchical agents within a flexible architecture.

Furthermore, we observe that complex tasks naturally decompose into a hierarchy of actions, which we term an **action pyramid**. For instance, the instruction to `obtain a diamond` can be broken down into skills like `gather wood` and `craft tools`, each further decomposing into grounding, motion, and raw actions. Inspired by this, we propose **long action-chain reasoning through action pyramid**, a framework that enables DeepHA to explicitly generate a sequence of linked abstract actions. In this process, higher-level actions serve as guiding thoughts for the generation of subsequent, more granular actions. This structured reasoning enhances decision-making; for example, generating `press(w)` is more reliably guided by an intermediate `move forward` thought than by a distant high-level objective alone. This framework also allows for dynamically scaling the depth of reasoning, and thus the computational effort, at inference time.

While powerful, the detailed action hierarchical process can impose substantial memory demands in long-horizon tasks, potentially leading to context lengths exceeding 600k tokens. To mitigate this, we develop a **memory-efficient hierarchical mechanism**. Recognizing that high-level abstracted actions persist across many low-level steps, this mechanism dynamically condenses the history of past low-level actions and redundant observations while preserving the active, high-level actions. By synergizing with the VLM's Key-Value (KV) caching, this approach achieves approximately a 75% reduction in context memory requirements without compromising performance.

We conduct extensive experiments in Minecraft, scaling existing benchmarks from under 100 to over 800 diverse tasks to rigorously evaluate generalization. Our results demonstrate that DeepHA, particularly when augmented with action hierarchy reasoning, establishes new state-of-the-art performance. Our main contributions are: (1) A unified Deep Hierarchical Agent (DeepHA) architecture that flexibly generates and executes multi-level, multi-modal abstract actions via a Mixture-of-Policies framework, bridging the VLA and hierarchical agent paradigms. (2) An action hierarchy reasoning framework that enables explicit, structured, and hierarchical decision-making by using higher-level actions as 'thoughts' for lower-level action prediction. (3) An efficient hierarchical-memory mechanism that significantly reduces (by 75%) the context memory footprint for long-horizon tasks by dynamically compressing history and leveraging KV caching. (4) The establishment of new state-of-the-art performance by DeepHA on an extensive benchmark of over 800 diverse Minecraft tasks, showcasing superior generalization performance.

## 2. Problem Formulation and Related Works

### 2.1. Action Spaces in LLM-based Agents

LLM-based agents typically generate actions in a two-stage process. First, an abstracted high-level description of the action is generated based on the task instruction and the current observation. We denote such descriptions as abstract actions $A$. Second, this abstract action is translated to a low-level, executable action of the environment. This process can be expressed as:

$$A \sim \pi_{LM}(\cdot \mid obs, ins), a \sim \pi_{policy}(\cdot \mid obs, A) \quad (1)$$

Here, $ins$ denotes the human-provided textual instruction, and $obs \in \mathbb{R}^{H \times W \times 3}$ represents the current visual observation. $A \sim \mathcal{A}$ signifies an *abstracted action* and is sampled from the predefined action space for LLM-based agents (Zhong et al., 2025a). The action space $\mathcal{A}$ ranges from language skills or sub-tasks, code, affordance, trajectory, goal state, latent representation, raw action, and reasoning (Liang et al., 2022; Driess et al., 2023; Belkhale et al., 2024a; Brohan et al., 2023; Zhen et al., 2024; Gu et al., 2023; Cai et al., 2023; Wang et al., 2024d; Bjorck et al., 2025). $A$ is generated by an autoregressive model, denoted $\pi_{LM}$, conditioned on both the $ins$ and the current $obs$, as shown in Eq. 1. Subsequently, $a$ represents a *low-level, discrete action* that the agent executes in the environment. These are often considered *raw actions*; $a$ could correspond to discrete keyboard inputs or mouse movements in computer interaction domains (Seed, 2025) and end controller position and forces in embodied robotics (Kim et al., 2024). The policy $\pi_{policy}$ (Eq. 1) is responsible for generating the action $a$, conditioned on the abstracted action $A$ and the current observation $obs$, which usually can be categorized into predefined MCP APIs (Xu & Peng, 2025), learning-based policies (Driess et al., 2023), or a rule-based parser (Team,

2025b).

This hierarchical formulation is general enough to encompass many recent Vision-Language-Action (VLA) models (Chi et al., 2023; Team et al., 2024b; Brohan et al., 2022a; 2023; Kim et al., 2024; Zheng et al., 2024; Zhang et al., 2024; Zhong et al., 2025b; Li et al., 2024b; Zhang et al., 2025; Zhu et al., 2025; Zhou et al., 2025; Chen et al., 2025). We conduct the experiments on the open-world Minecraft simulator, which has a series of hierarchical LLM-based agents and VLA models (Wang et al., 2023; Zhou et al., 2024; Fan et al., 2022; Deng et al., 2025; Zhao et al., 2024; Cai et al., 2023; Wang et al., 2023; 2024c;a; Li et al., 2024c; Cai et al., 2024c;b; Wang et al., 2024d; Zhou et al., 2024; Zheng et al., 2023).

## 2.2. Hierarchy between Action Spaces

We identify a distinct hierarchical structure within the action spaces of autonomous agents. This hierarchy is evident along two primary dimensions: temporal and semantic. Temporally, a single high-level action corresponds to a sequence of multiple low-level actions executed over time. Semantically, a high-level action functions as a conceptual abstraction, integrating a series of consecutive low-level actions into a single, meaningful command.

For example, in a gaming environment, the high-level motion command `go forward` is realized through a sequence of primitive, low-level actions, such as multiple `press(w)` keystrokes. This hierarchical principle extends beyond gaming to a wide range of applications for LLM-based agents. Consider a high-level command such as `search(query)`. Its execution is decomposed into a series of fundamental GUI operations: 1) opening a web browser, 2) typing the query into the search bar, and 3) pressing `Enter` to initiate the search.

It is crucial to distinguish this hierarchical action structure from planning paradigms such as the Tree of Thought (ToT) (Yao et al., 2023). Our action hierarchy is fundamentally a model of execution and abstraction, defining how a selected high-level intention is translated into a concrete sequence of low-level motor controls. In contrast, the Tree of Thought is a mechanism for deliberation and planning, which explores a branching tree of potential reasoning paths and action sequences to decide what the best overall strategy is. In essence, while ToT helps an agent decide which high-level goal to pursue from among multiple alternatives (e.g., "should I mine wood or find food?"), our action hierarchy defines the sub-steps required to accomplish the single goal once it has been chosen (e.g., "to mine wood, I must approach the tree, then swing the axe...").

The formation of this action hierarchy can be approached from two perspectives. A **top-down** approach involves pre-defining the action layers and employing a Large Language Model (LLM) for task decomposition and planning. In this paradigm, a complex, high-level action is systematically broken down into a sequence of simpler, executable sub-actions. Conversely, a **bottom-up** approach allows for the emergent formation of the hierarchy. Using algorithms inspired by Byte Pair Encoding (Sennrich et al., 2015) or other LLM-driven heuristics (Deng et al., 2025), the agent can automatically aggregate sequences of frequently co-occurring low-level actions to form abstract and composite actions.

We refer to this structured representation as an **action pyramid**. The detailed algorithms for the construction and utilization of this action pyramid are elaborated in the Appendix B.3.

## 3. Deep Hierarchical Agent

In this section, we present the Deep Hierarchical Agent (DeepHA). We first detail the model architecture in Section 3.1, which supports a mixture of heterogeneous action spaces. Next, in Section 3.2, we introduce an action hierarchy reasoning framework that scales inference-time computation, accompanied by three distinct inference modes. Finally, we describe a hierarchical memory mechanism in Section 3.3 that optimizes context management for long-horizon tasks.

### 3.1. Action Hierarchy within Agentic Models

Our model architecture is illustrated in Figure 2. Unlike previous LLM-based agents that are typically constrained to a monolithic abstract action space $A$ (as described in Equation 1), DeepHA is designed to generate actions from a **mixture of heterogeneous action spaces**, denoted as $\{\mathcal{A}_k\}$. This design enables the agent to flexibly select the most appropriate level of abstraction for a given task, enhancing both flexibility and specialization.

The core of our hierarchical agent consists of a high-level Vision Language Model (VLM), $\pi_{\text{LM}}$, and a low-level Mixture-of-Policies (MoP), $\{\pi_{\text{policy}}^k\}$. The VLM first generates a **typed abstract action** $A_k$, where $k$ indicates the chosen action space. A router then dispatches $A_k$ to the corresponding expert policy $\pi_{\text{policy}}^k$ for decoding into an executable, low-level action $a$. This two-stage generation process is formulated as:

$$(k, A_k) \sim \pi_{\text{LM}}(\cdot \mid o, \text{ins}), \quad a \sim \pi_{\text{policy}}^k(\cdot \mid o, A_k) \quad (2)$$

where $k \in \{1, \ldots, K\}$ serves as the routing token for the low-level policies.

In our implementation, the high-level VLM utilizes the Qwen2-VL architecture (Wang et al., 2024b). A key innovation is the model's ability to generate diverse abstract action types, each corresponding to a distinct ac-

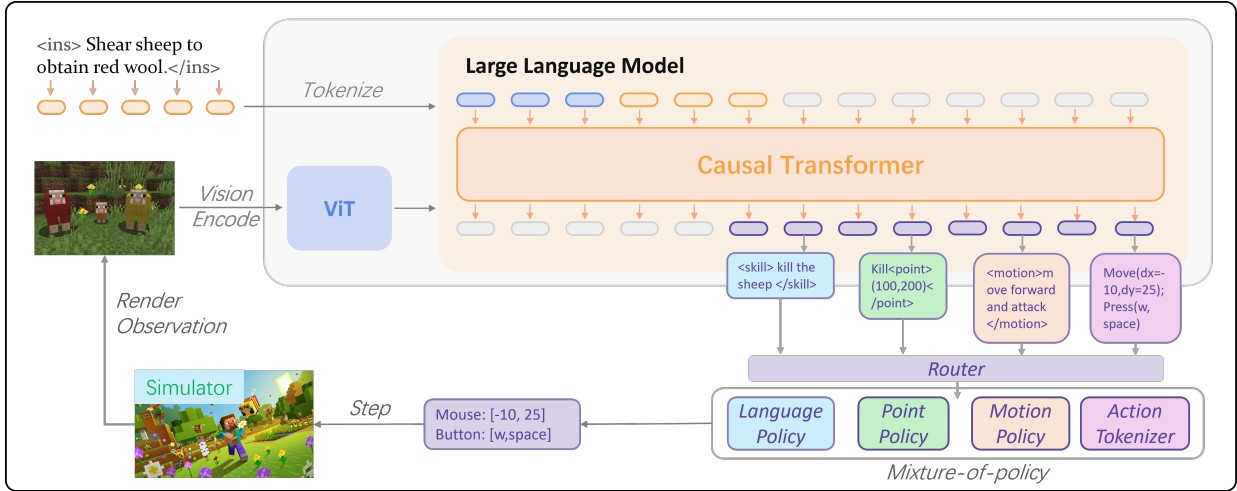

*Figure 2.* **Architecture of the Deep Hierarchical Agent (DeepHA).** Fed with the visual observations and language instructions, the vision language model autoregressively generates a hierarchical action chain, which can range from a high-level skill (e.g., `<skill>chop down a tree</skill>`) to a mid-level coordinate-based command (e.g., `<point>mine log (500, 500)</point>`) or a low-level motor action (e.g., `<motion>turn left</motion>`). A router dispatches the chosen action to a corresponding specialized policy within a Mixture-of-Policies module for execution in the simulator, thereby closing the agent-environment loop.

tion space $\mathcal{A}_k$. These include high-level language skills $A^s \in \mathcal{A}^s$ (e.g., `<skill>chop down a tree</skill>`), coordinate-based grounding actions $A^g \in \mathcal{A}^g$ (e.g., `<point>mine log (500, 500)</point>`), and direct motion commands $A^m \in \mathcal{A}^m$ (e.g., `<motion>turn left</motion>`). A comprehensive list of these action spaces is provided in Appendix B.3. To enable this, we repurpose reserved tokens in the VLM's vocabulary to represent distinct action types and fine-tune the model, a technique inspired by prior work (Brohan et al., 2023).

The **router** for the low-level policies is an implicit module that uses the index $k$ generated by the VLM to dynamically dispatch the abstract action $A_k$ to its corresponding expert. For instance, a motion command $A^m$ is routed to a dedicated motion policy $\pi^m_{\text{policy}}$. Each expert policy network employs a lightweight CNN and Transformer-XL architecture (Dai et al., 2019), following established practices (Lifshitz et al., 2024; Baker et al., 2022; Cai et al., 2024a). Further details on the policy network structures are available in Appendix A.

### 3.2. Scaling Action Hierarchies at Test-time

To enable end-to-end environmental interaction, we unify high-level planning and low-level execution within a single VLM. Rather than employing separate models, our VLM is trained to auto-regressively generate a sequence comprising both abstract thoughts and executable actions:

$$A_k, a \sim \pi_{\text{VLM}}(\cdot \mid \text{obs}, \text{ins}) \tag{3}$$

Here, the joint probability distribution is factorized as $\pi_{\text{LM}}(A_k \mid \text{obs}, \text{ins}) \cdot \pi_{\text{LM}}(a \mid A_k, \text{obs}, \text{ins})$. In this frame-

work, the abstract action $A_k$ serves as an intermediate "thought" that guides the generation of the final, executable action $a$. This end-to-end architecture simplifies the modeling process, allowing the agent to learn the mapping from high-level reasoning to low-level environmental interactions directly.

We further scale this formulation by employing an action hierarchy where higher-level actions act as contextual "thoughts" that guide lower-level generation. For instance, a motion command $A^m$ (e.g., "move forward") provides more direct guidance for a raw action $A^r$ (e.g., "press W") than a high-level skill $A^s$ alone. The VLM generates the full action hierarchy $(A^n, \ldots, A^1, a)$ auto-regressively. The joint probability is factorized as:

$$\begin{aligned} P(A^n, \ldots, A^1, a \mid \text{ins}, \text{obs}) &= p(A^n \mid \text{ins}, \text{obs}) \\ &\times p(A^{n-1} \mid A^n, \text{ins}, \text{obs}) \\ &\ldots \\ &\times p(A^1 \mid A^2, \ldots, A^n, \text{ins}, \text{obs}) \\ &\times p(a \mid A^1, \ldots, A^n, \text{ins}, \text{obs}) \end{aligned} \tag{4}$$

where each $p(\cdot)$ is modeled by the same VLM, conditioned on the instruction, observation, and all previously generated higher-level actions.

**Insights on Action Hierarchy as Thinking.** The integration of action hierarchy into the reasoning process fundamentally transforms the action prediction problem by effectively pruning the action space. Traditional agentic models often face a combinatorial explosion in environmental action spaces, where the dimensionality is the Cartesian product of all action components (e.g., $360 \times 360$ mouse movements

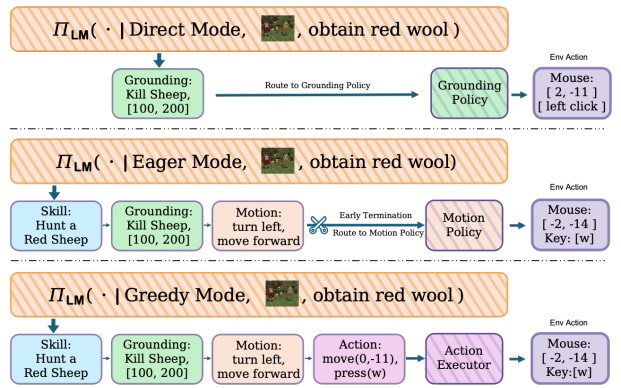

Figure 3. DeepHA inference modes. **Direct Mode** allows users to specify a single abstract action space. **Greedy Mode** generates actions step-by-step as thoughts, ultimately producing raw actions. **Eager Mode** truncates the chain based on the Greedy process and uses a specialized policy to decode the selected abstract action.

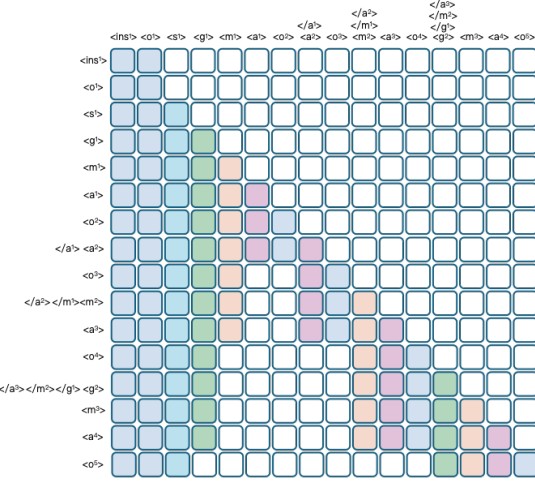

Figure 4. Illustration of the DeepHA inference process with the efficient hierarchical memory mechanism. The figure shows an example attention pattern over the token sequence, illustrating how different token groups attend to historical context during inference. Filled cells indicate visible tokens, while hollow cells denote masked positions. : Initial instruction. <o>: Visual observation. , <g>, <m>, <a>: Tokens for skill, grounding, motion, and raw actions, with superscripts indicating action sequence IDs. </g>, </m>, </a>: Stop tokens marking the action completion.

multiplied by $2^N$ key combinations). This results in a search space exceeding 16 million possibilities, leading to high-entropy predictions. By introducing hierarchical "thoughts," we systematically reduce this dimensionality. For example, a decision to "move camera left" eliminates all dimensions associated with rightward movement ($dx > 0$), allowing the model to focus on the magnitude of the shift; similarly, "move forward" rules out backward-related keys. This hierarchical decomposition converts a massive, flat action space into a sequence of tractable, low-entropy decisions, significantly enhancing prediction accuracy. Furthermore, unlike two-stage approaches that may suffer from cascading errors, treating high-level actions as internal reasoning steps preserves access to the complete action space. We draw a parallel here to the "aha moments" in chain-of-thought reasoning (Guo et al., 2025); in low-level control, these moments correspond to the selection of pivotal high-level actions that dramatically simplify the subsequent decision-making process.

Based on this framework, DeepHA supports three inference modes, categorized by whether the VLM uses an action chain for thinking and whether it utilizes the Mixture-of-Policies to decode actions (see Appendix D for case studies):

**Direct Mode.** In this mode, the VLM generates an action $A_k$ from a single, pre-specified abstract action space $\mathcal{A}_k$, which can be manually set via a system prompt. This approach minimizes the VLM's per-step computational load by focusing generation on a single level of abstraction, offering the flexibility to switch between action types (e.g., executing raw actions $a$ for crafting vs. motion commands $A^m$ for navigation).

**Greedy Mode.** This mode executes the complete top-down action hierarchy reasoning process. The VLM sequentially generates the full action hierarchy (e.g., $A^s \rightarrow A^g \rightarrow$

$A^m \rightarrow a$). Each higher-level action serves as an explicit thinking step guiding the generation of the subsequent, more concrete action. While this ensures the most thorough reasoning, it is computationally intensive as the full chain is generated at every decision step.

**Eager Mode.** This mode balances reasoning depth and computational cost by enabling early termination of the generation chain. It initiates the hierarchical thinking process as in Greedy Mode (e.g., generating $A^s$ and $A^g$), but halts generation once an executable intermediate action (e.g., $A^m$) is produced. This allows the agent to benefit from high-level thinking while avoiding the full computational overhead. We employ two termination mechanisms: (1) **Manually-Controlled Termination:** Halts generation upon detecting a pre-specified action tag (e.g., `</motion>`), primarily used for controlled ablation studies. (2) **Learned Dynamic Termination:** To grant full autonomy, the VLM is fine-tuned to generate a special `<|Eager-Stop|>` token when it determines the current intermediate action is sufficient for execution, allowing dynamic adaptation of reasoning depth.

### 3.3. Action Hierarchy Leads to Hierarchical Memory

While the deep reasoning of the Greedy Mode enhances performance, its sequential generation of the full action hierarchies at each step leads to a rapid expansion of the VLM's context length in long-horizon tasks. This can exceed hundreds of thousands of tokens, significantly impairing the model's reasoning capabilities due to issues like "lost in

*Table 1.* Evaluation results of Minecraft agents on over 800 tasks. For each task category, we report the success rate on a representative task (indicated by the icon), the best-of-10 successful tasks within the category (*Bo10*), and the average success rate across all tasks in that category (*ASR* $\pm$ standard deviation). Results highlighted in blue represent the second-best performances, while those in red indicate the state-of-the-art performance for each metric across all agents. '-' signifies tasks where the agent failed to achieve any success.

| Method | Mine Blocks | | | | Kill Entities | | | | Craft Items | | | | Smelt Items | | | |
|---|---|---|---|---|---|---|---|---|---|---|---|---|---|---|---|---|
| | ⬛↑ | *Steps*↓ | *Bo10*↑ | *ASR*↑ | 🟥 | *Steps*↓ | *Bo10*↑ | *ASR*↑ | ↗ | *Steps*↓ | *Bo10*↑ | *ASR*↑ | 🪨 | *Steps*↓ | *Bo10*↑ | *ASR*↑ |
| *Instruction-Conditioned Policies* | | | | | | | | | | | | | | | | |
| VPT (Baker et al., 2022) | 20 | 377 | 30.7 | $6.0^{\pm11.4}$ | 10 | 396 | 24.6 | $3.6^{\pm7.7}$ | 0 | 398 | 6.7 | $0.8^{\pm3.3}$ | - | - | - | - |
| STEVE-I (Lifshitz et al., 2024) | 50 | 384 | 29.4 | $8.0^{\pm17.0}$ | 0 | 395 | 14.7 | $3.9^{\pm12.0}$ | 0 | 391 | 16.4 | $3.2^{\pm8.4}$ | - | - | - | - |
| ROCKET-1 (Cai et al., 2024a) | 60 | 392 | 57.5 | $18.9^{\pm24.3}$ | 60 | 320 | 63.9 | $27.9^{\pm29.3}$ | - | - | - | - | - | - | - | - |
| JARVIS-VLA (Li et al., 2025) | 55 | 305 | 55.3 | $30.0^{\pm35.4}$ | 60 | 352 | 61.9 | $18.5^{\pm22.7}$ | 40 | 339 | 74.3 | $25.1^{\pm23.9}$ | 60 | 387 | 92.3 | $65.4^{\pm32.3}$ |
| *Hierarchical Agents* | | | | | | | | | | | | | | | | |
| LanguageHA (Driess et al., 2023) | 60 | 365 | 31.3 | $11.3^{\pm14.5}$ | 0 | 393 | 12.8 | $6.5^{\pm9.3}$ | 5 | 397 | 19.3 | $6.3^{\pm9.2}$ | - | - | - | - |
| PointHA (Cai et al., 2024a) | 90 | 290 | 61.0 | $37.1^{\pm38.5}$ | 50 | 346 | 90.1 | $26.5^{\pm23.4}$ | 15 | 380 | 27.5 | $6.7^{\pm10.8}$ | 10 | 402 | 37.0 | $5.0^{\pm8.7}$ |
| MotionHA (Belkhale et al., 2024b) | 70 | 336 | 51.0 | $27.4^{\pm35.2}$ | 20 | 392 | 29.5 | $4.3^{\pm10.8}$ | - | - | - | - | - | - | - | - |
| LatentHA (Wang et al., 2024d) | 70 | 363 | 54.2 | $24.4^{\pm31.1}$ | 50 | 371 | 24.6 | $8.5^{\pm17.9}$ | 0 | 393 | 19.1 | $3.0^{\pm7.5}$ | 0 | 437 | 2.2 | $0.2^{\pm1.5}$ |
| OpenHA (Wang et al., 2025a) | 80 | 287 | 67.3 | $30.1^{\pm13.9}$ | 70 | 316 | 62.6 | $32.5^{\pm9.2}$ | 80 | 314 | 58.8 | $31.9^{\pm13.7}$ | 75 | 339 | 92.3 | $66.6^{\pm28.1}$ |
| *Ours* | | | | | | | | | | | | | | | | |
| DeepHA (SFT) | 100 | 269 | 82.4 | $48.9^{\pm43.6}$ | 90 | 307 | 92.3 | $38.9^{\pm45.3}$ | 75 | 317 | 85.6 | $43.2^{\pm26.5}$ | 90 | 293 | 93.5 | $82.4^{\pm22.6}$ |
| DeepHA (RL) | 100 | 248 | 86.2 | $62.3^{\pm47.9}$ | 100 | 293 | 93.4 | $45.4^{\pm31.2}$ | 95 | 297 | 92.3 | $51.9^{\pm25.2}$ | 100 | 305 | 93.5 | $88.6^{\pm32.6}$ |

the middle," where critical instructions are overlooked (Li et al., 2024a; Liu et al., 2025). Furthermore, the context becomes saturated with redundant observations and low-level actions from consecutive steps where the environment changes minimally (Seed, 2025).

To address this, we introduce an efficient hierarchical-memory mechanism that dynamically manages the VLM context. As illustrated in Figure 4, this approach leverages the hierarchical nature of tasks, where high-level abstract actions persist over multiple low-level steps. The process unfolds as follows: 1) **Initial Generation and Re-use:** The agent initially generates a complete action hierarchy chain, establishing a high-level action (e.g., $A_1^g$) as a contextual anchor. This high-level action is reused across subsequent steps while the agent generates the necessary low-level actions (e.g., $a_1, a_2, \ldots$). 2) **Memory Compression:** Upon the completion of an intermediate action, signaled by a stop token (e.g., `</g>`), our mechanism triggers a memory compression step. It prunes the detailed execution history—including the intermediate actions and their corresponding visual observations (depicted as shaded tokens in Figure 4)—from the input sequence, while preserving the high-level semantic goal. Consequently, the agent maintains a concise yet informative context, enabling efficient long-horizon reasoning. And this dynamic context management yields significant efficiency gains by synergizing with the VLM.

**Efficient Context Inference with KV Caching.** By pruning tokens from the input sequence, we drastically reduce the computational load for the subsequent generation step. Critically, the computational influence of the pruned history is preserved in the VLM's Key-Value (KV) cache. This strategy avoids the costly re-computation (prefilling) of past

states, allowing the agent to benefit from a long-term history without the full cost of reprocessing it at every step.

**Training with Causal Attention Masks.** This inference-time process informs our training methodology. We employ causal attention masks to train the model to understand that for an ongoing high-level goal, it should attend to the initial goal definition rather than the full, detailed history of already-completed sub-tasks.

## 4. Experiments and Discussions

### 4.1. Experimental Setups

**Simulator and Benchmarks.** We employ Minecraft (Version 1.16.5) as our primary test environment (Guss et al., 2019). The agent's observation space consists solely of first-person RGB visual images, with a resolution of $360 \times 640 \times 3$. The action space comprises discretized, human-like mouse and keyboard controls. Specifically, it includes: mouse displacement, mouse clicks and keyboard inputs. Further detailed specifications of the observation and action spaces are provided in Appendix C.1. We utilized OpenHA's benchmarks (Wang et al., 2025a) to evaluate our agent model, which includes over 800 tasks across embodied mineblock, GUI crafting, and combat tasks, among others. Our primary metrics are **Success Rate** and **Best-of-N accuracy** across tasks, and average steps to finish the task. In our final analysis, we report the average metrics aggregated across all tasks within each of the three groups.

**Baselines.** We evaluate our proposed method against two main categories of baseline agents. Instruction-Conditioned Policies: This group includes representative models such as OpenAI VPT (Baker et al., 2022), the

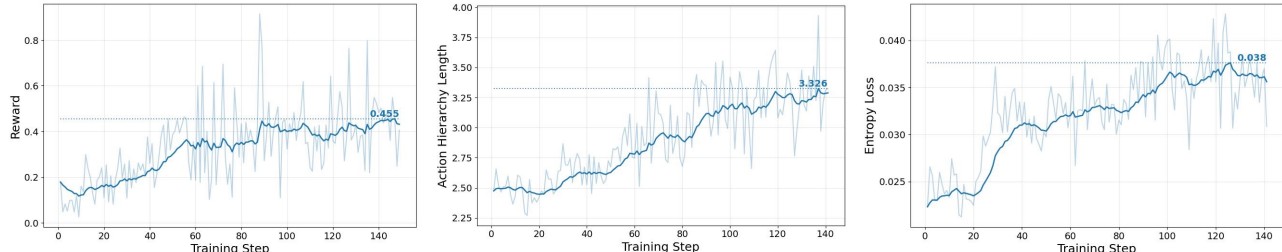

*Figure 5.* DeepHA increases with the training steps in the RL process, leading to a higher success rate in task completion and a deeper action hierarchy in thought responses.

language-conditioned STEVE-I (Lifshitz et al., 2024), the visual-prompt-conditioned ROCKET-1 (Cai et al., 2024a), and the Vision-Language-Action (VLA) model JARVIS-VLA (Li et al., 2025). These methods typically encode the input instruction into an embedding and directly predict actions. Hierarchical Agents: This category consists of agents with a two-level structure, where a high-level Vision-Language Model (VLM) processes instructions and observations to generate an abstract intermediate action (e.g., a language skill, visual coordinates, motion primitive, or latent code), which is then executed by a low-level policy. We consider four representative agents according to the modality of the abstract action: LanguageHA (Driess et al., 2023) and MotionHA (Belkhale et al., 2024b), which use fine-tuned Qwen2-VL-7B (Wang et al., 2024b) as the high-level VLM paired with specialized low-level policies (Baker et al., 2022); PointHA (Cai et al., 2024a), which employs Qwen2-VL-7B as the high-level VLM with ROCKET-1 (Cai et al., 2024a) as its low-level policy; and LatentHA, operating within the OmniJARVIS framework (Wang et al., 2024d), which similarly uses Qwen2-VL-7B as its high-level component with FSQ-GROOT (Van Den Oord et al., 2017) as its low-level policy (Cai et al., 2024c). Our comparisons benchmark performance against both flat policy structures and contemporary hierarchical approaches.

**Training Recipes.** We employ a two-stage training pipeline to equip DeepHA with both fundamental capabilities and strategic planning actions. Initially, we perform **Supervised Fine-Tuning (SFT)** on a multi-space expert dataset to provide a cold start, enabling the model to generate valid actions across varying levels of abstraction. To further empower the agent with autonomous exploration and robust long-chain action hierarchy thinking, we subsequently utilize **Multi-Turn Reinforcement Learning (MTRL)** with a variant of Group Relative Policy Optimization (GRPO) (Shao et al., 2024). In this phase, the agent interacts with the environment to generate trajectories $\tau$, optimizing the policy $\pi_\theta$ to maximize the episodic success rate while minimizing generation costs. The objective function is defined as:

$$J(\theta) = \mathbb{E}_{x\sim\mathcal{T}}\mathbb{E}_{\tau\sim\pi_\theta(\cdot|x)}\left[\mathbb{1}\{\text{success}(\tau)\} - \lambda l_\theta(\tau)\right] \quad (5)$$

where $\mathbb{1}\{\text{success}(\tau)\}$ represents the binary reward for task

completion, and $l_\theta(\tau)$ serves as a length penalty to encourage the model to prefer concise action spaces when possible. This approach effectively transitions the agent from imitation to autonomous reinforcement, fostering robust performance in complex open-world scenarios.

### 4.2. Main Results

To rigorously assess generalization, we expanded the evaluation benchmark tenfold from 80 to over 800 distinct tasks, categorized into MineBlocks, KillEntities, CraftItems, and SmeltItems (in Appendix C.2). This challenging suite reveals the brittleness of prior methods; for instance, ROCKET-1 (Cai et al., 2024a), previously state-of-the-art with a 93% success rate on its original mining benchmark, achieves only 18.9% ASR on our expanded set. To better measure capability breadth, we also introduce a new metric, BoN accuracy, which quantifies the percentage of unique tasks an agent can complete at least once within N trials.

As presented in Table 1, our DeepHA establishes a new state-of-the-art, demonstrating superior performance and generalization. The key insight from these results is that DeepHA consistently surpasses not only generalist models but also the strongest specialized baselines in their respective domains. This validates our core hypothesis that an agent capable of dynamically operating across a mixture of action spaces is fundamentally more robust and capable.

Specifically, in MineBlocks and KillEntities group, tasks that favor precise spatial reasoning, DeepHA outperforms the specialized PointHA by significant margins in ASR (+25.2 and +18.9 points, respectively). This demonstrates that DeepHA's flexible architecture can adopt a point-centric strategy when optimal, while enhancing it with deeper reasoning. The advantage is even more pronounced in complex, multi-step categories. In CraftItems and SmeltItems GUI tasks, which require long-horizon planning, DeepHA achieves an ASR of 51.9% and 88.6%, substantially exceeding the next-best hierarchical agent, OpenHA, by over 20 and 22 points.

These results confirm that by transcending the limitations

*Table 2.* Ablation experiments on action chain depth revealed that as the chain deepens, the agent's performance consistently improves when utilizing the same outcome action.

| Outcome | Depth | Action Chain | ASR(%) ↑ |
|---|---|---|---|
| Raw action | 1 | Raw | $30.0^{\pm 35.4}$ |
| Raw action | 2 | Motion → Raw | $39.2^{\pm 31.7}$ |
| Raw action | 2 | Grounding → Raw | $42.6^{\pm 39.3}$ |
| Raw action | 3 | Grounding → Motion → Raw | $48.9^{\pm 43.6}$ |

*Table 3.* DeepHA with efficient memory achieves a 3x reduction in memory tokens while maintaining performance comparable to full memory.

| Method | Images | Tokens ↓ | ASR ↑ | Bo10 ↑ |
|---|---|---|---|---|
| No Context Memory (Markovian) | 1 | 385 | 37.1 | 68.3 |
| Full Memory (Sliding Window) | 20 | 7948 | 38.5 | 78.5 |
| Efficient Memory | 5 | 1976 | 39.6 | 77.3 |

*Table 4.* Comparison of inference modes across different outcome actions. The maximum context token length is set as 8k.

| Mode | Action | Embodied ↑ | GUI ↑ | FPS ↑ | Context ↓ |
|---|---|---|---|---|---|
| Direct | Motion | $32.9^{\pm 38.7}$ | $2.7^{\pm 8.6}$ | 6.7 | 8000 |
| Eager | Motion | $37.6^{\pm 33.3}$ | $\mathbf{11.5^{\pm 13.6}}$ | 4.9 | **1934** |
| Direct | Grounding | $37.7^{\pm 35.8}$ | $8.0^{\pm 11.4}$ | 6.9 | 8000 |
| Eager | Grounding | $\mathbf{40.7^{\pm 28.3}}$ | $\mathbf{23.4^{\pm 19.6}}$ | 6.4 | **1926** |
| Direct | Raw | $34.5^{\pm 31.9}$ | $26.6^{\pm 25.1}$ | 2.3 | 8000 |
| Greedy | Raw | $\mathbf{48.9^{\pm 43.6}}$ | $\mathbf{43.2^{\pm 26.5}}$ | 1.2 | **1976** |

of a single, fixed action modality, DeepHA achieves a new level of generalization. Its ability to dynamically select the appropriate level of action abstraction—from precise coordinates for mining to high-level skills for crafting—allows it to excel across a far wider variety of tasks than any single-paradigm agent.

## 4.3. Analysis and Discussions

### 4.3.1. ABLATION ON ACTION HIERARCHY DEPTH

We conducted a controlled ablation to quantitatively measure how performance varies with the depth of the action hierarchy. To ensure a fair comparison and isolate the effect of action depth itself, the final outcome action is always a Raw Action. The only variable is the length and composition of the preceding action chain generated before producing the final output. Results are shown in Table 2.

Directly generating a Raw Action with depth 1 serves as the baseline, achieving an ASR of 30.0%. Increasing the action depth to 2 significantly boosts performance: incorporating a Motion action raises ASR to 39.2%, while using a more informative Grounding action increases it further to 42.6%. Finally, the full action chain (Grounding → Motion → Raw) achieves the best performance, reaching 48.9%.

This step-by-step improvement provides strong quantitative evidence for our central claim: greater action depth directly correlates with higher task success rates. Each additional level in the action hierarchy supplies valuable context that refines the final decision, validating the core motivation for our architecture and the principle of scaling inference-time computation for more effective deliberation.

### 4.3.2. EXPERIMENT ON THE HIERARCHICAL MEMORY

In this section, we examine how our efficient hierarchical memory mechanism affects both the performance and efficiency of DeepHA. Specifically, we evaluate the proposed memory mechanism by analyzing its impact on average context length, average success rates, and Bo10 score. The comparison is conducted across three configurations: a minimal-history baseline DeepHA (No Memory), a version retaining full uncompressed trajectory history (Full Memory), and our proposed trajectory-compressed variant (Efficient Memory).

The results in Table 3 compellingly demonstrate the benefits of our approach. DeepHA (Efficient Memory) reduces the average context length to 1976 tokens from 7948 in the Full Memory setting. Importantly, this significant context reduction is achieved not only without performance degradation but with a slight improvement in ASR: DeepHA (Efficient Memory) achieves 39.6%, surpassing both Full Memory (38.5%) and No Memory (37.1%). Furthermore, its Bo10 accuracy of 77.3% is nearly on par with Full Memory (78.5%) and markedly better than No Memory (68.3%).

These findings indicate that our mechanism not only yields substantial computational efficiency—implying faster inference and lower resource usage—but also enhances agent scalability for long-horizon tasks while maintaining, or even improving, task success. This improvement stems from more effective memory management: redundant low-level details are pruned, while strategically relevant high-level information is preserved, enabling more focused reasoning.

### 4.3.3. COMPARISONS OF INFERENCE MODES

We conduct experiments to evaluate the inference speed and performance across different inference modes within DeepHA. The experimental results are shown in Table 4.

We find that Eager Mode outperforms Direct Mode based on the same outcome action space. The partial reasoning in Eager Mode provides a clear performance benefit. For example, Eager (Motion) achieves 37.6% ASR versus 32.9% from Direct (Motion), proving the value of the initial action hierarchy "thoughts". Eager Mode offers an excellent performance/efficiency trade-off compared to Greedy Mode: Eager Mode is substantially more efficient, operating up to 5x faster than Greedy Mode on inference speed (e.g., 6.4 FPS vs. 1.2 FPS). It achieves this speed while retaining a large portion of Greedy Mode's performance (e.g., 40.7%

ASR vs. 48.9% ASR for Eager (Grounding) vs. Greedy (Raw)), making it a highly practical choice. These results quantitatively demonstrate that Eager Mode successfully balances computational cost with high performance.

## 5. Conclusions

We propose Deep Hierarchical Agent, a novel vision-language-action model that advances instruction-following in open-world Minecraft. It integrates a flexible, scalable action hierarchy reasoning mechanism for dynamic control modality selection and adaptive reasoning depth, and an efficient hierarchical design that cuts context memory for long-horizon tasks without performance loss. We also identify high-level Vision-Language Model interaction frequency as a key new scaling dimension for hierarchical agents, and develop fast/slow reasoning modes to dynamically manage inference-time computation. DeepHA achieves state-of-the-art performance and provides a powerful, computationally adaptable framework for open-world agent development.

## Impact Statement

This paper presents work whose goal is to advance the field of Machine Learning. There are many potential societal consequences of our work, none which we feel must be specifically highlighted here.

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

# A. DHA Model Details

## A.1. Vision Language Models

The core of our DHA's high-level reasoning and multi-modal understanding is a pre-trained Vision-Language Model (VLM). For this, we utilize **Qwen2-VL-7B** (Wang et al., 2024b) as our base model. We conduct post-training on this VLM to adapt it to the Minecraft domain and the task of generating hierarchical actions, but we make no fundamental modifications to its underlying architecture (e.g., its Transformer layers or vision encoder structure). This VLM is responsible for processing both textual instructions (e.g., user commands) and visual observations (i.e., first-person perspective images from the Minecraft environment) to produce a selected abstracted action.

Inspired by the methodology presented in RT-2 (Brohan et al., 2023), we integrate action generation capabilities into the VLM by re-purposing a subset of existing, typically infrequently used, tokens within its language tokenizer's vocabulary. These tokens are endowed with specific semantics corresponding to environment-related actions or structured action components. This approach allows the VLM to output diverse types of abstracted actions (as detailed in Section **??**) directly within its generative output space, without requiring changes to its core vocabulary size or output head architecture. The VLM thus learns to map complex multi-modal inputs to these specialized action tokens as part of the hierarchical decision-making process.

## A.2. Mixture of Policies

Given that the high-level VLM in DHA can generate various distinct types of abstracted actions (i.e., language skills $A^s$, grounding actions $A^g$, motion actions $A^m$, and raw action specifications $A^r$), a flexible mechanism is required to translate these diverse abstractions into concrete, environment-executable actions. To achieve this, we employ a **Mixture of Policies (MoP)** architecture for the low-level control module, which conceptually resembles a Mixture-of-Experts (MoE) system.

This MoP module consists of multiple specialized low-level policy networks, each an "expert" in handling a specific type of abstracted action. These expert policies are structured in parallel. A crucial component of this design is a **Router module**. When the high-level VLM generates an abstracted action, the Router inspects this output (typically by identifying the type or category of the generated abstracted action, e.g., via special tokens indicating if it's an $A^s$, $A^g$, $A^m$, or $A^r$) and dispatches it to the corresponding expert policy for further processing and execution.

Specifically, each low-level expert policy within the MoP module shares a common architectural foundation but can be further specialized:

- **Core Architecture:** The policies are generally based on the **Transformer-XL** (Dai et al., 2019) architecture, enabling them to handle sequences and maintain memory where necessary.

- **Visual Processing:** Each policy that requires visual input (e.g., for grounding or fine-grained control based on the current view) incorporates an **Impala CNN encoder** (Baker et al., 2022) to process the raw visual image observations ($obs_t$) and convert them into fixed-dimensional visual embeddings. These embeddings are then provided as input to the Transformer-XL policy body.

- **Parameter Size:** Each individual expert policy model has approximately **100 million parameters**.

- **Specializations and Foundations:**
    - The **Grounding-based Policy** (for $A^g$) is designed based on principles from **Rocket-1** (Cai et al., 2024a), likely adept at interpreting coordinate-based instructions or visual targets and translating them into navigational actions.
    - The **Motion-based Policy** (for $A^m$) and the **Language Skill-based Policy** (for $A^s$) draw inspiration and architectural elements from **VPT (Video PreTraining)** (Baker et al., 2022), suggesting they are well-suited for translating sequential or language-defined skills and motion commands into more concrete action sequences.
    - The **Raw Action Policy** (often referred to as an Action Tokenizer in our diagrams, for $A^r$) is responsible for translating structured raw action specifications from the VLM (e.g., descriptions of mouse movements and key presses) into the discrete keyboard and mouse operations permissible by the Minecraft environment.

This MoP architecture allows DHA to leverage the strengths of different action abstractions and specialized low-level controllers, contributing to its versatility and effectiveness across a wide range of tasks.

# B. Details of Training Datasets

## B.1. Detailed Description of Base Datasets for Minecraft World Knowledge

To address the disparity in Minecraft-specific world knowledge between open-source Vision-Language Models (VLMs) and leading proprietary models (e.g., the GPT series, Claude series), we have developed a comprehensive suite of base datasets. This initiative is driven by compelling experimental evidence that indicates deep, domain-specific world knowledge is crucial for achieving high Vision-Language-Action (VLA) performance in Minecraft (Li et al., 2025; Wang et al., 2024d). Our primary objective at this stage is to significantly improve the foundational Minecraft understanding and capabilities of open-source VLMs. The dataset, which primarily originates from and is further expanded based on resources such as JARVIS-VLA (Li et al., 2025), encompasses four main categories detailed below. The generation and curation of these components extensively utilize self-instruct methodologies and leverage the capabilities of powerful foundation models:

- **World Knowledge Question Answering (QA):** This component aims to instill a strong textual understanding of the Minecraft world. We initiated this by crawling a comprehensive textual corpus from the Minecraft Wiki. Utilizing this corpus, a set of approximately 202K questions was generated via a self-instruct methodology, employing large language model (LLM) such as GPT-3.5-turbo(Ouyang et al., 2022). Subsequently, diverse and high-quality answers to these questions were generated by advanced LLMs, including GPT-4o (Hurst et al., 2024), Claude-3.5 Sonnet (Anthropic, 2025), and Gemini 1.5 Pro (Team et al., 2024a), operating in a zero-shot setting. The resulting 277K QA pairs were then subjected to an automated quality control process. In this step, a VLM, augmented with access to the relevant Minecraft Wiki texts, evaluated and filtered the answers to ensure accuracy and relevance.

- **Visual Question Answering (VQA) and Scene Captioning:** To enrich the model's visually grounded knowledge, we curated a diverse collection of approximately 35K images. These images were carefully selected from video recordings of human gameplay in Minecraft, ensuring a wide variety of scenes and contexts. GPT-4o (Hurst et al., 2024) was then employed to generate detailed captions for these images. Furthermore, GPT-4o engaged in a self-interrogation process (i.e., asking and answering questions) based on the visual content of each image to create a rich set of VQA pairs, yielding approximately 15K image-caption pairs and 20K VQA instances.

- **Visual Grounding:** For fine-grained object and region understanding within Minecraft scenes, we constructed a dedicated visual grounding dataset. An automated annotation algorithm was first employed to propose candidate grounding points and bounding boxes for salient objects or regions in approximately 600K Minecraft images. The accuracy of these initial (often noisy) proposals was then critically assessed and refined by GPT-4o (Hurst et al., 2024), which served as a visual critic. This process resulted in a high-fidelity dataset of 404K visual grounding annotations.

- **Reasoning Capabilities Enhancement:** To specifically bolster the model's capacity for in-game reasoning and planning, we adopted a two-pronged strategy. Firstly, we incorporated the Open-Thoughts dataset (Team, 2025a) (comprising 75K instances) to warm-up our pre-trained VLM, thereby fostering its latent reasoning abilities. Secondly, we designed a novel set of 5K unique Minecraft-centric tasks that necessitate multi-step logical deduction or planning to complete. The warmed-up VLM was then tasked with generating explicit reasoning pathways (chains of thought) for these tasks. A rule-based critic was subsequently employed, along with rejection sampling, to curate and select only high-quality and correct reasoning chains.

In total, this aggregated dataset, comprising approximately 300k pairs, was utilized for Supervised Fine-Tuning (SFT) of our Qwen2-VL based model (as detailed in Section 3.1). The objective of this SFT stage is to equip the model with robust foundational knowledge of Minecraft and a preliminary yet effective capacity for exploration and task completion within its environment.

## B.2. Mixture of Abstracted Action Prediction Datasets

After processing the base instruction-following trajectories to derive datasets for each type of abstracted action (i.e., language skills $A^s$, grounding actions $A^g$, motion actions $A^m$, and raw actions $A^r$, we combine these into a comprehensive "mixture of abstracted action" dataset. This composite dataset is pivotal for training our Deep Hierarchical Agent (DHA) and is structured into two primary components, serving different stages of our training pipeline (see Section E).

**1. Markovian Abstracted Action Data.** This component of the dataset is designed primarily for the initial pre-training of the Vision-Language Model (VLM) to accurately generate various individual abstracted actions. Each instance in this

dataset follows a Markovian structure, typically consisting of a triplet: $(\text{ins}_t, \text{obs}_t, A_t^*)$ where $\text{ins}_t$ is the high-level language instruction, $\text{obs}_t$ is the current visual observation, and $A_t^*$ is the target abstracted action to be predicted at timestep $t$. $A_t^*$ can be an instance of a language skill $(A^s)$, a grounding action $(A^g)$, a motion action $(A^m)$, or a raw action sequence $(A^r)$.

Crucially, to guide the VLM in generating the correct type and format of action, each training instance is accompanied by a specific system prompt. This prompt explicitly instructs the model on the desired format for its response and specifies the permissible action space for that particular instance. For example, a prompt might ask the VLM to generate a 'motion action' or a 'raw action sequence' based on the input.

When constructing this part of the dataset, we mix data instances corresponding to different abstracted action types. Specifically, for the non-language abstracted actions, we maintain a sampling ratio where instances requiring grounding actions $(A^g)$, motion actions $(A^m)$, and raw actions $(A^r)$ are mixed in a ratio of approximately **1:4:16**, respectively. This ratio is designed to provide sufficient exposure to the more frequently occurring and granular raw actions while ensuring the model also learns higher-level motion and grounding primitives. Instances involving language skill $(A^s)$ generation are also included in this mixture, though their sampling might follow a different distribution or be determined by their natural occurrence in the processed trajectories. This dataset component primarily facilitates Stage 1 of our training pipeline (Action Prediction Pre-training), enabling the VLM to master the semantics and generation of diverse, individual action abstractions.

**2. Action Hierarchy Structured Data.** The second component of our dataset is formatted specifically for action hierarchy learning. This data is crucial for the post-training phase (Stage 2) where the DHA learns to perform hierarchical reasoning by sequentially generating linked actions. Each instance in this dataset represents a complete abstracted action chain for a given observation $\text{obs}_t$ and instruction $\text{ins}_t$, formulated as: $(\text{ins}_t, \text{obs}_t, (A_t^s, A_t^g, A_t^m, A_t^r))$ This structure explicitly provides the full hierarchy of actions, from the high-level language skill $A_t^s$ down to the executable raw action $A_t^r$, with each intermediate action $(A_t^g, A_t^m)$ conditioned on its predecessors in the chain, as detailed by our action hierarchy formulation. Training on this hierarchy-structured data enables the DHA to learn the dependencies between different levels of action abstraction and to use higher-level actions as guiding "thoughts" for generating subsequent, more detailed actions. This phase is critical for imbuing the agent with its advanced reasoning and planning capabilities.

By training on this two-part mixture dataset, DHA first learns to proficiently generate individual types of abstracted actions in a prompted manner (via the Markovian data) and then learns to strategically chain these actions together for complex, multi-step task execution (via the action hierarchy structured data).

## B.3. Action Spaces and Action Pyramid

**Grounding.** Grounding actions involve using visual coordinates to specify the location of objects instead of directly naming them. By combining a symbolic verb (e.g., `mine`) with object coordinates (e.g., `coordinate=[180, 320]`), grounding actions decouple the "what" from the "where." This enables a policy to execute a skill on any object by targeting its location.

**Motion.** These consist of sequences of continuous actions that involve physical movement or changes in position. Examples include commands such as `move forward`, `turn right`, `jump` and compositions of short motion actions. Each motion action typically involves fine-grained control over the agent's movement dynamics, requiring precise, low-level policies that adjust parameters such as speed, direction, and acceleration. These actions are often dependent on the agent's current environment and may require real-time adjustments to account for obstacles or changes in the surroundings. The execution of motion actions necessitates a continuous flow of feedback and adjustments to ensure the agent's movement aligns with the intended goal.

**Raw action.** Low-level, direct control actions, such as keyboard and mouse events, that correspond to basic human-like interactions with the environment. These actions involve discrete or continuous inputs, such as moving the mouse or pressing keys, providing immediate control over the agent's behavior in real time.

**Action Pyramid.** As illustrated in Section 2.2, different action types in Minecraft exhibit a naturally hierarchical structure. High-level actions can often be decomposed into a sequence of lower-level ones. For instance, a skill action such as *chop down tree* can be broken into grounding actions, including *approach the tree* located at coordinates (320, 270), followed by *mine the log* at coordinates (500, 500). Each grounding action can in turn be realized by a sequence of motion actions or raw motor actions, such as discrete keyboard and mouse events. When constructing our datasets across different abstraction levels, we explicitly record the mapping between each abstract action and its corresponding low-level motor actions. This alignment allows us to track not only which abstract operation is executed, but also where and how it is grounded in the environment. The positional information (e.g., screen coordinates of targets) provides a concrete link between symbolic

instructions and perceptual states, enabling more robust training of grounding policies. Furthermore, by exploiting this hierarchical mapping, we can connect actions across multiple abstraction levels and form coherent action chains. These hierarchical chains naturally stack together, thereby forming an **action pyramid** that unifies skill, grounding, motion, and raw actions together

## C. Simulator and Evaluation Tasks

### C.1. Observation Space and Action Space

We adopt native Minecraft-rendered images as the observation space for all agents. The action space is defined by human-like mouse and keyboard controls. Mouse displacement is discretized into 1800 bins. When the agent is in a GUI interface, these displacements correspond to cursor movements; otherwise, they adjust the camera view by changing pitch ($\Delta y$) and yaw ($\Delta x$). Mouse clicks are represented by dedicated tokens for left, right, and middle button presses. Keyboard inputs are encoded as unique tokens corresponding to alphabetic characters, numeric digits, and a predefined set of special keys (e.g., `Shift`, `Space`, `Escape`, `Enter`, etc.). A detailed mapping of mouse clicks and keyboard inputs is provided in Table 5.

*Table 5.* The environmental action space we use in Minecraft that can step the environment directly.

| Index | Action | Human Action | Description |
|---|---|---|---|
| 1 | Forward | key W | Move forward. |
| 2 | Back | key S | Move backward. |
| 3 | Left | key A | Strafe left. |
| 4 | Right | key D | Strafe right. |
| 5 | Jump | key Space | Jump. When swimming, keeps the player afloat. |
| 6 | Sneak | key left Shift | Slowly move in the current direction of movement. |
| 7 | Sprint | key left Ctrl | Move quickly in the direction of current motion. |
| 8 | Attack | left Button | Destroy blocks (hold down); Attack entity (click once). |
| 9 | Use | right Button | Interact with the block. |
| 10 | hotbar.[1-9] | keys 1 - 9 | Selects the appropriate hotbar item. |
| 11 | Yaw | move Mouse X | Turning; aiming; camera movement. Ranging from -180 to +180. |
| 12 | Pitch | move Mouse Y | Turning; aiming; camera movement. Ranging from -180 to +180. |
| 13 | Equip | - | Equip the item in main hand from inventory. |
| 14 | Craft | - | Execute a crafting recipe to obtain new item. |
| 15 | Smelt | - | Execute a smelting recipe to obtain new item. |

### C.2. Evaluation Tasks

We evaluate the performance of our method on a diverse suite of tasks within the Minecraft environment, categorized into five main groups: mining blocks, crafting items, killing entities, smelting items, and interacting with blocks. To thoroughly assess generalization capabilities, we have designed an extensive benchmark comprising over 800 distinct tasks distributed across these groups, with each group containing at least 100 different task variations.

Task success is determined automatically by parsing information returned by the environment upon specific events. For example, if the agent is given a task with the natural language description such as "use an iron pickaxe to mine a diamond ore," successful completion is logged if the environment registers events corresponding to the criteria for that task (e.g., an event like `mine_block:diamond_ore` occurring, potentially with conditions on tool usage like `use_item:iron_pickaxe` if specified in the success criteria for that particular task variant). It is important to note that this backend success-checking information is utilized exclusively for evaluation and is not available to the agent as part of its observation space, which consists solely of visual images.

Below, we describe the general setup for one of the major task categories, "mine blocks," as an illustration of our task design methodology.

**Task Setup for *Mine Blocks*.** Tasks within the "Mine Blocks" category require the agent to locate and mine a specific type of block within the Minecraft world. Each individual task, identified by a unique name (e.g., `mine_block:oak_log`, `mine_block:iron_ore`), is procedurally generated with controlled initial conditions to ensure reproducibility and varied challenges. The setup for each task variant is defined by the following parameters, as exemplified by the JSON structure

you've provided:

- **Target Block**: The specific Minecraft item ID of the block to be mined (e.g., `oak_log`, `diamond_ore`).

- **Initial Conditions** (`seeds`): Each task variant is associated with one or more predefined world `seed` values and specific agent starting `position` coordinates (x, y, z). This allows for multiple scenarios for mining the same block type, varying the environmental context and agent starting location. For example, the task `mine_block:diamond_ore` might have multiple seed/position combinations for increased diversity.

- **Permitted/Required Tools** (`tool`): A list of item IDs for tools that are either required or permitted for successfully mining the target block. This can range from a single specific tool (e.g., `["diamond_pickaxe"]` for `obsidian`) to a list of acceptable tools of varying materials (e.g., `["stone_pickaxe", "iron_pickaxe", "diamond_pickaxe"]` for `iron_ore`). Some tasks may even specify tools with particular enchantments, such as `wooden_shovel:{Enchantments:[{id:silk_touch,lvl:1}]}` for mining `grass_block` to obtain the block itself rather than dirt.

For instance, a task to mine iron ore might be defined as `mine_block:iron_ore`, with an initial setup placing the agent at coordinates like `[194, 61, 897]` within a world generated by seed 2025. Successful completion would require the agent to mine an iron ore block, permissibly using a stone, iron, or diamond pickaxe. The automatic evaluation would check for the corresponding `mine_block:iron_ore` event from the environment, under valid tool conditions if specified.

**Task Setup for *Craft Items*.** The "Craft Items" task category challenges the agent to create specific items using the in-game crafting system. Each task, such as `craft_item:dispenser` or `craft_item:black_dye`, sets a specific `goal` item for the agent to produce. The setup for these tasks is defined by several key parameters derived from your JSON data:

- **Target Item** (`goal`): The unique item ID of the object the agent must craft (e.g., `dispenser`, `golden_leggings`).

- **Initial Inventory** (`init_inventory`): A predefined set of items and materials provided to the agent at the start of the task. Each entry specifies the item `type` (e.g., `redstone`, `cobblestone`, `wither_rose`), the required `quantity` (which can be exact or a minimum, e.g., `">=7"`), and often a `slot` for initial placement (e.g., `"random"`). This ensures the agent has the necessary, and sometimes exact, raw materials for the recipe.

- **Crafting Table Requirement** (`need_crafting_table`): A boolean value (`true` or `false`) indicating whether the target item's recipe necessitates the use of a crafting table, or if it can be crafted directly in the player's 2x2 inventory grid.

- **Initial Conditions** (`seeds`): Similar to other task categories, each crafting task variant can be associated with specific world `seed`(s) and agent starting `position` coordinates, although the initial inventory is typically the more critical factor for task success in this category.

For example, the task `craft_item:dispenser` requires the agent to craft a dispenser. The agent would start with an initial inventory containing at least seven cobblestone blocks, one redstone dust, and one bow, and would need to use a crafting table. In contrast, a task like `craft_item:black_dye` can be completed without a crafting table, starting with at least one wither rose. Task success is determined automatically by checking if the agent's inventory contains the specified `goal` item after its actions, based on information logged by the environment.

## D. Case Study

We list a few cases of different inference modes in Figure 6. We present several observations from rollout examples for the tasks `kill a sheep` in eager mode, `craft paper` in greedy mode, and `chop trees` in direct mode.

## E. Training Recipe

Our Deep Hierarchical Agent (DHA) is trained via a multi-stage pipeline designed to progressively build its capabilities, from foundational knowledge to sophisticated chained-action generation. Key hyperparameters for these three stages are summarized in Table 6.

Eager Mode

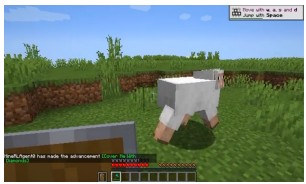 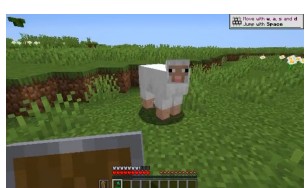 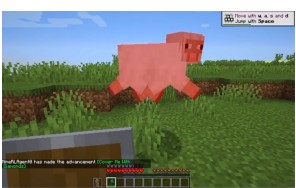

The Skill Action is <skill>kill a sheep</skill>. The Grounding Action is <grounding> <|object_ref_start|>sheep<|object_ref_en d|><|point_start|>(500, 560)<|point_end|> </grounding>. The Motion Action is <motion>turn right and attack</motion>. The Raw Action is <raw> move(0.6, 0), press(w) and click(left)</raw>.

The Skill Action is <skill>kill a sheep</skill>. The Grounding Action is <grounding> <|object_ref_start|>sheep<|object_ref_en d|><|point_start|>(500, 500)<|point_end|> </grounding>. The Motion Action is <motion>go forward and attack</motion>. The Raw Action is <raw>press(w) and click(left)</raw>.

The Skill Action is <skill>kill a sheep</skill>. The Grounding Action is <grounding> <|object_ref_start|>sheep<|object_ref_en d|><|point_start|>(500, 500)<|point_end|> </grounding>. The Motion Action is <motion>go forward and attack</motion>. The Raw Action is <raw>press(w) and click(left)</raw>.

The Skill Action is <skill>kill a sheep</skill>. The Grounding Action is <grounding> <|object_ref_start|>sheep<|object_ref_en d|><|point_start|>(460, 500)<|point_end|> </grounding>. The Motion Action is <motion>attack</motion>. The Raw Action is <raw>click(left)</raw>.

Greedy Mode

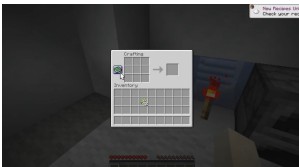 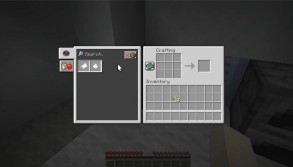 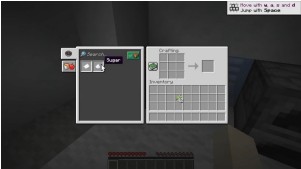 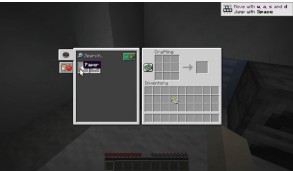

The Skill Action is <skill>craft paper</skill>. The Grounding Action is <grounding> <|object_ref_start|>recipe book<|object_ref_end|><|point_start|>(3 94,352)<|point_end|> </grounding>. The Motion Action is <motion>left click</motion>.

The Skill Action is <skill>craft paper</skill>. The Grounding Action is <grounding> <|object_ref_start|>paper<|object_ref_en d|><|point_start|>(282,339)<|point_end| > </grounding>.

The Skill Action is <skill>craft paper</skill>. The Grounding Action is <grounding> <|object_ref_start|>paper<|object_ref_en d|><|point_start|>(282,339)<|point_end| > </grounding>.

The Skill Action is <skill>craft paper</skill>. The Grounding Action is <grounding> <|object_ref_start|>paper<|object_ref_en d|><|point_start|>(282,339)<|point_end| > </grounding>. The Motion Action is <motion>left click</motion>

Direct Mode

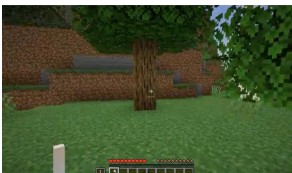 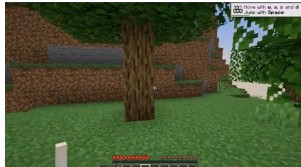 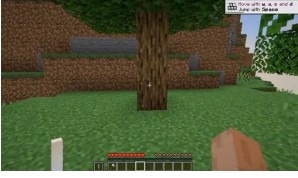 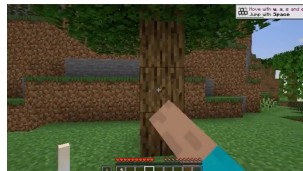

The Raw Action is <raw>press(w) and click(left)</raw>.

The Raw Action is <raw>move(-1, 0) and click(left)</raw>.

The Raw Action is <raw>press(w) and click(left)</raw>.

The Raw Action is <raw>click(left)</raw>.

*Figure 6.* Case study of inference modes.

The training proceeds as follows:

**Stage 0: Basic Knowledge Finetuning.** First, we imbue the base Vision-Language Model (VLM) with fundamental Minecraft domain knowledge. This involves supervised finetuning of the LLM backbone using 3 billion tokens from our curated basic knowledge datasets.

**Stage 1: Mixture of Abstracted Action Prediction Pre-training.** Next, all components of the VLM are pretrained on a significantly larger dataset (11 million images) comprising a mixture of abstracted actions. This stage teaches the model the semantics of diverse action types (e.g., language actions, grounding actions, motion actions, and raw actions) and how to associate them with instructions and observations, without yet enforcing explicit sequential dependencies.

**Stage 2: Action Hierarchy Post-Training.** Finally, the entire DHA is post-trained using 1 billion tokens of data formatted specifically for action hierarchy learning. This teaches the model to generate actions in a sequentially dependent, hierarchical manner, where higher-level actions guide lower-level ones, refining its reasoning and planning capabilities. This three-stage pipeline ensures our DHA first acquires essential domain knowledge, then learns a broad repertoire of actions, and ultimately masters the sophisticated action hierarchical reasoning process for complex task execution.

*Table 6.* Training setup and hyperparameters for the three training stages of our DHA.

| Stages | Stage 0 | Stage 1 | Stage 2 |
|---|---|---|---|
| Training budget (tokens) | 3B | 100B | 1B |
| Training Images | 550K | 11M | 319K |
| Trainable Components | LLM backbone | all | all |
| Batch Sizes | 128 | 512 | 128 |
| LR warmup steps | 100 | 200 | 100 |
| Maximum LR | $5.0 \times 10^{-5}$ | $5.0 \times 10^{-6}$ | $3.0 \times 10^{-6}$ |
| Minimum LR | $5.0 \times 10^{-6}$ | 0 | 0 |

