# OpenReview forum: "DeepHA: Scaling Action Chains Elicits Deep Hierarchical Agents"
_ICML.cc/2026/Conference — ICML 2026 regular_

### Official Review · Reviewer_pPTo · 2026-02-21

**Soundness:** 3
**Presentation:** 2
**Significance:** 3
**Originality:** 3
**Overall Recommendation:** 4
**Confidence:** 2

**Summary:**

This paper proposes DeepHA, a hierarchical vision-language-action agent that dynamically generates multi-level action chains, from high-level skills to low-level motor actions. The core idea is to treat higher-level actions as structured “thoughts” to guide lower-level execution. The authors also introduce an efficient hierarchical memory mechanism to reduce long context length. The method is evaluated on more than 800 Minecraft tasks and shows improved performance compared to several existing baselines.

**Compliance With Llm Reviewing Policy:**

Affirmed.

**Final Justification:**

The author’s response addressed most of my concerns. However, given the significant difference in our professional backgrounds, I am unable to assess the novelty of this paper. That said, it appears to me that a considerable amount of work effort went into this paper, so I recommend that the AC, SAC, and PC base their decisions primarily on the comments from other reviewers. For now, I am assigning a “weak accept” rating.

**Key Questions For Authors:**

Questions:

1.	The explanation of the abstract action space remains somewhat unclear. Although the author provides textual details in Appendix B3, I still recommend that the author provide an example for each action type, ideally with detailed action data examples or images for intuitive description.
2.	Can the author provide comparison results between state-of-the-art world model-based methods (e.g., DreamerV3) and lifelong learning-based methods (e.g., Voyager) in Minecraft?
3.	How sensitive is the performance to the hierarchy depth and termination strategy? Is the improvement mainly due to deeper reasoning, or simply due to increased computation budget at test time?

**Limitations:**

No. The author should include a limitation section.

**Strengths And Weaknesses:**

Strength:

1.	The paper proposes a clear hierarchical design and unifies multiple action abstractions into one framework, which is conceptually interesting.
2.	The action-chain reasoning idea is intuitive and supported by ablation studies showing deeper chains lead to better performance.
3.	The large-scale evaluation (800+ tasks) is comprehensive and demonstrates strong performance improvements within the chosen benchmark.

Weakness:

1.	The comparison method is limited to hierarchical agents, lacking comparative results for other approaches, which casts doubt on the fairness of the findings and the validity of the conclusions.
2.	The article's methodology appears to be applicable only within the Minecraft gaming environment. This simulated setting lacks real-world relevance, raising questions about whether its purported advanced capabilities are merely overfitting results specific to Minecraft. Whether it can be extended to other domains, such as embodied AI, remains unknown.
3.	Although the hierarchical reasoning idea is well motivated, the paper could better clarify which component contributes most to the final performance gain. For example, the relative contribution of action depth, memory compression, and RL fine-tuning is not fully disentangled, making it harder to understand where the main improvement comes from.

---

> ### Author Rebuttal · Authors · 2026-03-31
>
> > W1: *"Comparison limited to hierarchical agents, lacking results for other approaches."*
>
> We thank the reviewer for raising this concern. Our comparison is not restricted to hierarchical agents. In Table 1, we include both policy/VLA baselines (VPT, STEVE-1, ROCKET-1, JARVIS-VLA) and hierarchical baselines (LanguageHA, PointHA, MotionHA, LatentHA, OpenHA), and we will further clarify this positioning in the revision.
>
> > W2: *"Methodology appears applicable only within Minecraft... extensibility to other domains remains unknown."*
>
> To directly address this, we have conducted new experiments on Language Table (Lynch et al., 2023), a continuous 2-DoF robotic control task. All models were trained on under 1B tokens with Qwen2-VL-7B:
>
> | Task | VLA | HA(Motion) | HA(Grounding) | DeepHA |
> |:-:|:-:|:-:|:-:|:-:|
> | BlockToBlock | 83±9 | 76±8 | 82±7 | **92±4** |
> | BlockToRelLoc | 82±5 | 83±7 | 85±5 | **93±5** |
> | SeparateBlocks | 96±2 | 98±1 | 94±2 | **98±2** |
>
> DeepHA achieves consistent improvements (+9 over VLA on BlockToBlock, +11 on BlockToRelLoc), confirming that the contribution is the reasoning-and-routing framework itself, not a Minecraft-specific design.
>
> > W3: *"Relative contribution of action depth, memory, and RL is not fully disentangled."*
>
> We thank the reviewer for this suggestion. In DeepHA, action depth drives the core performance gain, hierarchical memory makes this deep reasoning practical at scale, and RL enables the model to autonomously discover the optimal reasoning depth.
>
> Deeper action chains progressively constrain the raw action space. Table 2 shows this: with the final output fixed to Raw Action, ASR improves from 30.0 (depth 1) to 42.6 (depth 2, Grounding) to 48.9 (depth 3), and more informative constraints yield larger gains (Grounding 42.6 > Motion 39.2).
>
> However, deep reasoning over long-horizon tasks can exceed 600K tokens of context. Hierarchical memory compresses completed low-level history while preserving active high-level goals via KV-cache reuse, reducing average context from 7,948 to 1,976 tokens (~75%) without performance loss (Table 3).
>
> Multi-turn RL further enables the agent to autonomously discover how deep to reason. After RL, the model spontaneously generates deeper chains (Figure 5) and achieves a substantial further gain (Mine Blocks ASR: 48.9 → 62.3, Table 1).
>
> > Q1: *"Abstract action space explanation remains unclear... please provide an example for each action type."*
>
> Thank you for this suggestion. We provide concrete examples below and will add visual illustrations in the appendix:
>
> | Level | Action Type | Example |
> |---|---|---|
> | 4 | Language Skill | `<skill>kill a sheep</skill>` |
> | 3 | Grounding Action | `Grounding: Combat(object='sheep', position=[551, 500])` |
> | 2 | Motion Action | `Motion: wave the sword.` |
> | 1 | Raw Action | `mouseMove(10,-1), click(left), press(w)` |
>
> In an action chain, each higher-level action serves as a structured constraint that guides the generation of the next lower-level action.
>
> > Q2: *"Can the author provide comparison results between DreamerV3 and Voyager?"*
>
> | Method | Embodied | Combat | Craft |
> |---|--:|--:|--:|
> | DreamerV3 | 42.3 | 34.3 | N/A |
> | DeepHA (Ours) | 86.2 | 93.4 | 92.3 |
>
> DreamerV3 cannot complete Craft tasks because it is trained in MineDojo, which does not support GUI-based interaction required by our Craft setting. We will add DreamerV3 as an additional baseline in the revision.
>
> For Voyager, the setting differs substantially: it uses a code-based action space and privileged observations beyond first-person RGB, whereas ours is restricted to visual observations and keyboard-and-mouse control. A direct numerical comparison would not be fair, and we will discuss Voyager qualitatively in the revision.
>
> > Q3: *"How sensitive is the performance to hierarchy depth and termination strategy — deeper reasoning or simply more test-time computation?"*
>
> This is an important question. Our results suggest that the improvement is not merely due to spending more computation at test time, but due to using that computation in a more structured way through action hierarchy reasoning. In Table 2, when the final output is fixed to the same raw action space, increasing the hierarchy depth consistently improves performance, indicating that the extra reasoning steps provide more informative intermediate constraints for action prediction. In Table 4, the comparison between Direct, Eager, and Greedy further shows that termination strategy matters: Eager achieves stronger performance than Direct under the same outcome action space, while also being much more efficient than Greedy. This indicates that the benefit comes from better action-space search and controllable reasoning depth, rather than from computation alone. We will make this distinction clearer in the revision.

---

> > ### Author Rebuttal · Reviewer_pPTo · 2026-03-31
> >
> > I thank the Authors for their response. I have no further questions, and I will keep my score unchanged.

---

### Official Review · Reviewer_i1tq · 2026-02-23

**Soundness:** 2
**Presentation:** 3
**Significance:** 3
**Originality:** 3
**Overall Recommendation:** 3
**Confidence:** 4

**Summary:**

The paper introduces the Deep Hierarchical Agent (DeepHA), a unified Vision-Language Model (VLM) architecture designed to overcome the limitations of decoupled planning and execution in autonomous agents. Rather than maintaining separate modules for high-level strategy and low-level control, DeepHA operates over a unified, heterogeneous action space. It formulates decision-making as an autoregressive generation task where the model can output both high-level semantic skills (e.g., "chop tree") and low-level motor controls (e.g., specific mouse and keyboard actions) within the same sequence.

Central to this approach is the "action hierarchy reasoning" framework, where generated high-level actions serve as structured "thoughts" or context that condition the subsequent generation of precise low-level motor sequences. To mitigate the severe memory constraints caused by predicting extremely long sequences of low-level actions, the authors integrate KV cache compression techniques.  The model is trained via a comprehensive three-stage pipeline consisting of domain knowledge acquisition, massive-scale action learning (utilizing 100B tokens), and hierarchical reasoning fine-tuning. Extensive experiments demonstrate that DeepHA achieves state-of-the-art performance in the highly complex and open-ended Minecraft environment.

**Compliance With Llm Reviewing Policy:**

Affirmed.

**Final Justification:**

After further consideration alongside the other reviewers' feedback, I am lowering my score from 4 to 3. The rebuttal confirmed that the core mechanism remains a domain-specific hierarchical decomposition of a Minecraft action space, with novelty residing more in systems engineering than in a generalizable algorithmic insight. This limitation is compounded by the prohibitive reproduction cost of 4,352 A800 GPU-hours and the fact that cross-domain validation was restricted to a relatively simple 2-DoF Language Table task. Taken together, the work reads as a well-executed but heavily resource-bound system refinement, and a score of 3 more accurately reflects this overall assessment.

**Key Questions For Authors:**

1. Cross-Domain Generalizability and Sample Efficiency: The Stage 1 training requires a massive 100B tokens and 11M images, which appear highly tailored to the Minecraft environment. Can the authors provide theoretical insights, or ideally, small-scale empirical evidence (e.g., in a simpler robotics simulator), on how effectively this "mixture of action spaces" paradigm transfers to other domains? A convincing explanation or preliminary data showing that the architecture does not strictly require 100B domain-specific tokens to work in new environments would significantly increase my confidence in the method's broad applicability, solidifying or potentially raising my score.

2. Error Recovery in Autoregressive Generation: Because the model relies on autoregressive generation for both high-level thoughts and low-level execution, it is inherently susceptible to compounding errors. If the agent executes a faulty sequence of low-level motor controls (e.g., missing the target due to visual occlusion), how does the single-stream VLM detect this failure and trigger a "re-planning" of the high-level semantic skill?Addressing this will resolve my primary concern regarding the robustness and technical soundness of using a single unconstrained sequence generation process for closed-loop control.

3. Reproducibility and Artifact Release: Given the immense computational budget required for Stage 0 and Stage 1, reproducing this work is computationally prohibitive for the vast majority of academic labs. Do the authors commit to open-sourcing the pre-trained model weights, the curated hierarchical reasoning dataset (Stage 2), or providing a scaled-down "recipe" for researchers with limited compute? A clear commitment to open science and artifact release would completely alleviate my concerns regarding the paper's presentation and long-term impact on the community.

**Limitations:**

No. While the authors have presented an impressive technical achievement, the submission would be significantly strengthened by a more upfront and detailed discussion of its limitations and potential negative societal impacts.

1. Computational and Environmental Costs: The authors should explicitly quantify the computational resources (e.g., total GPU hours) and the estimated carbon footprint required for the massive three-stage training pipeline (especially the 100B token phase). Acknowledging the barrier to entry this creates for the broader research community is important.

2. Robustness and Safety: The authors should candidly discuss the limitations of purely autoregressive control in safety-critical scenarios. In real-world deployments (unlike Minecraft), a single hallucinated or incorrect low-level token could lead to catastrophic physical hardware damage.

3. Societal Impact of Generalist Agents: As this architecture brings the field closer to highly capable, autonomous generalist agents that can bridge reasoning and computer control, the authors should briefly touch upon the dual-use nature of their work (e.g., the potential for such agents to be repurposed for malicious automated tasks or cyber-attacks if deployed as web/computer agents).

**Strengths And Weaknesses:**

1. Soundness

Strengths: The methodology is technically rigorous and well-justified. Formulating hierarchical control purely as an autoregressive sequence generation problem over a mixed vocabulary is an elegant solution to the planner-actor disconnect. Furthermore, the incorporation of KV cache compression is not merely an engineering trick but a vital, sound systemic design choice that makes long-horizon low-level control computationally feasible for VLMs. The empirical results in Minecraft are robust, and the ablation of the training stages provides strong support for the proposed architecture.

Weaknesses: Because the agent relies heavily on autoregressive generation for long sequences of low-level actions, it is inherently susceptible to compounding errors (e.g., a slight hallucination in spatial grounding leading to a failed sequence of mouse clicks). The submission would be technically sounder if it provided a more thorough empirical analysis of error-recovery mechanisms—specifically, how the single-stream model detects execution failures and correctly aborts a low-level sequence to generate a new high-level corrective skill.

2. Presentation

Strengths: The submission is clearly written, well-structured, and easy to follow. The motivation—highlighting the rigidity of decoupled symmetric systems—logically sets up the need for a unified mixture of action spaces. The detailed breakdown of the three-stage training pipeline (Table 6) provides excellent clarity on how the model acquires its capabilities step-by-step.

Weaknesses: While the methodology is clearly explained, the sheer scale of the experiments (e.g., Stage 1 requiring 100B tokens and 11M images with all parameters trainable) raises concerns about reproducibility for the broader academic community. To improve the presentation, the authors should explicitly discuss the compute requirements and clarify whether model checkpoints, data, or a scaled-down training recipe will be open-sourced to allow expert readers to reproduce or build upon the results.

3. Significance

Strengths: Bridging the gap between abstract, long-horizon reasoning and precise, high-frequency motor control is one of the most critical bottlenecks in embodied AI and robotics today. By demonstrating that a single scaled VLM can effectively manage this hierarchy without falling back on hard-coded scripts, this paper addresses a highly relevant problem. The impressive SOTA performance in Minecraft proves the practical utility of the method. This paradigm is highly likely to influence future research in generalist robotic agents and large action models (LAMs).

4. Originality

Strengths: The work demonstrates strong originality through a creative combination and repurposing of existing ideas. While using LLMs/VLMs for control or employing Chain-of-Thought reasoning is not new, translating the "thought" directly into a high-level action anchor that autoregressively dictates a stream of low-level motor tokens within a single vocabulary is a highly novel perspective. It successfully removes the restrictive assumptions of prior hierarchical RL methods that rely on separated value functions or isolated sub-policies, offering fresh insights into how Scaling Laws might be applied directly to complex action spaces.

---

> ### Author Rebuttal · Authors · 2026-03-31
>
> > W1: *"Stage 1 training requires a massive 100B tokens... Can the authors provide how effectively this paradigm transfers to other domains?"*
>
> We first apologize for the confusion: the Stage 1 token count is a typo in the current draft and should read **10B** rather than 100B.
>
> More importantly, the core idea of DeepHA—composing typed action spaces into a progressive constraint hierarchy and scaling inference-time reasoning depth—is not tied to large Minecraft-specific pretraining. In many agent domains, multiple action abstractions already exist (trajectories, keypoints, motion primitives, semantic skills); our contribution is to organize them into an action pyramid and let the model reason through progressively deeper chains.
>
> To support this empirically, we have conducted new experiments on Language Table (Lynch et al., 2023), a continuous 2-DoF robotic control task unrelated to Minecraft. All models were trained on **under 1B tokens** with Qwen2-VL-7B:
>
> | Task | VLA | HA(Motion) | HA(Grounding) | DeepHA |
> |:-:|:-:|:-:|:-:|:-:|
> | BlockToBlock | 83±9 | 76±8 | 82±7 | **92±4** |
> | BlockToRelLoc | 82±5 | 83±7 | 85±5 | **93±5** |
> | SeparateBlocks | 96±2 | 98±1 | 94±2 | **98±2** |
>
> DeepHA achieves consistent improvements across all tasks (+9 over VLA on BlockToBlock, +11 on BlockToRelLoc), demonstrating that the architecture does not inherently require extremely large domain-specific pretraining to be effective in a new environment. The progressive-pruning paradigm transfers once task-appropriate action interfaces are provided.
>
> > W2 + Q2: *"How does the VLM detect failure and trigger re-planning? "*
>
> We agree that error recovery and safety are crucial concerns for autoregressive control. DeepHA is not a one-shot sequence generator that commits to a long action plan without feedback; instead, it operates in a non-Markovian closed loop. At each decision step, the VLM conditions on the current observation together with the retained interaction history and previously generated higher-level actions, then generates a new action chain. If a low-level execution fails—for example due to occlusion or inaccurate motor control—the subsequent observation reflects this mismatch and the model revises its next generated hierarchy accordingly. Re-planning is thus achieved by iterative regeneration under updated observations rather than by a separate external planner. Our hierarchical memory mechanism further supports this by preserving the active high-level goal while compressing completed low-level details, helping the model maintain long-horizon intent while responding to new evidence.
>
> That said, we do not claim that a purely autoregressive controller is sufficient for safety-critical real-world deployment. The framework does offer a partial mitigation: before producing the final raw action, the model emits interpretable intermediate actions (grounding targets, motion-level intentions), making the agent's intended behavior more transparent and enabling earlier intervention by a human supervisor or safety monitor. However, for real-world deployments, additional safety layers—such as execution constraints, formal low-level safety controllers, and human oversight—would be necessary. We will add this discussion explicitly in the revision.
>
> > W3 + Q1: *"Can the authors quantify computational costs and commit to open-sourcing weights, datasets, or a scaled-down recipe?"*
>
> We agree that computational cost should be reported more explicitly. As noted in W1, the Stage 1 token count is a typo (10B, not 100B). Based on Table 6, all experiments were conducted on 32 NVIDIA A800-SXM4-80GB GPUs with the following budgets:
>
> | Stage | Tokens | Images | Trainable | Compute |
> |---|---|---|---|---|
> | Stage 0 | 3B | 550K | LLM backbone | 768 A800 GPU-hrs |
> | Stage 1 | 10B | 11M | all modules | 3,200 A800 GPU-hrs |
> | Stage 2 | 1B | 319K | all modules | 384 A800 GPU-hrs |
>
> The total cost is 4,352 A800 GPU-hours across all stages. We acknowledge this may create a barrier for many academic labs. To address this, we plan to open-source the full data-processing pipeline, training and evaluation code, curated Stage 2 datasets, and trained model checkpoints. We will also provide a scaled-down training recipe—for example, applying LoRA to the backbone and training only the hierarchy-reasoning and routing modules on Stage 2 data—so that researchers can reproduce the core findings without matching the full pretraining budget.
>
> > Q3: *"The authors should briefly touch upon the dual-use nature of their work."*
>
> We appreciate this suggestion and agree that the broader-impact discussion should be expanded. While our experiments are conducted in Minecraft, the method could have dual-use implications if adapted to computer-agent settings. Potential risks include malicious automation, unauthorized task execution. In the revision, we will add a concise discussion of these risks in the limitations.

---

> > ### Author Rebuttal · Reviewer_i1tq · 2026-04-03
> >
> > Thanks for your response. I have no further questions.

---

### Official Review · Reviewer_hNyV · 2026-03-13

**Soundness:** 3
**Presentation:** 3
**Significance:** 3
**Originality:** 2
**Overall Recommendation:** 4
**Confidence:** 4

**Summary:**

This paper presents DeepHA, a hierarchical agent for Minecraft that operates over multiple action spaces including skill, visual grounding, motion, raw actions, rather than fixed abstraction. The method combines a high-level VLM with a mixture of specialized low-level policies, and introduce a Chain-of-Action (CoA) reasoning process. The authors further introduce a hierarchical memory mechanism that compress past low-level execution history while keeping compact higher-level semantic context. Experiments on a large Minecraft benchmark with 800+ tasks show strong performance over prior baselines.

**Compliance With Llm Reviewing Policy:**

Affirmed.

**Final Justification:**

My primary concerns were: (1) limited novelty over OpenHA, (2) dependence on hand-designed, Minecraft-specific hierarchy, and (3) insufficient implementation detail. The rebuttal addressed all three adequately. The depth ablations convincingly show that progressive action-space pruning is a distinct contribution from OpenHA's action-space selection. The new Language Table experiments demonstrate cross-domain transferability, and the authors committed to releasing code, data, and checkpoints.
The paper still leans more toward a strong systems contribution than a cleanly novel method, and action spaces remain manually defined. However, the empirical results are solid, the ablations are well-designed, and the cross-domain evidence strengthens the generality claim. I raise my score from 3 to 4 (weak accept).

**Key Questions For Authors:**

* Could the authors provide a direct and clear comparison listing the difference relative to OpenHA?
* How is the action hierarchy constructed in practice? How many supervision is required to create the hierarchy-related training data?
* How general is this approach beyond Minecraft? How much efforts would it required to be applied or transferred to other domains?

**Limitations:**

No. The authors should discuss limitations more explicitly, especially the dependence on domain-specific action design and data curation and the limited evidence outside Minecraft.

**Strengths And Weaknesses:**

**Strengths**

* Hierarchical action space is important for agent to efficient planning and compositional generalization in embodied spaces.  This paper addresses an important problem, which is well motivated that no single action space is optimal for all tasks.

* The overall system is coherent and competent. The combination of a high-level VLM, multiple specialized policies, hierarchical action generation is reasonable and the empirical results on 800+ tasks are strong. The ablation is useful, suggesting that deeper action chains can achieve better performance than shallow ones.

* The proposed hierarchical memory and different work modes are efficient and flexible for tasks of different difficulties and horizons.

**Weaknesses**

* My main concern is the originality relative to closely related prior work, especially OpenHA. The mixed action spaces, CoA reasoning and evaluation on a large Minecraft benchmark is very close to the OpenHA's central narrative. This paper contributes a more diverse hierarchy action space and reads like a system refinement than a clearly differentiated new method.

* Another concern is that the method appears heavily dependent on hand-designed hierarchy and domain-specific data construction. This makes it difficult to judge how general the method is beyond Minecraft.

* The paper would benefit from a clearer presentation, including more details about implementations, action hierarchy design, data curation pipeline.

---

> ### Author Rebuttal · Authors · 2026-03-31
>
> > W1: *"Reads like a system refinement rather than a differentiated method."*
>
> We thank the reviewer for this important question. DeepHA does share the Minecraft benchmark and multiple action spaces, but the two papers ask different questions and arrive at different methods.
>
> OpenHA asks *"which action space is best?"* and trains an All-in-One agent on a mixture with a fixed two-level hierarchy. While DeepHA asks *"how should an agent handle the combinatorial explosion of a 16M+ raw action space?"* Rather than selecting among action spaces, DeepHA composes them as complementary constraint layers—each level (Skill → Grounding → Motion → Raw) progressively prunes the search space for the next. The ablations confirm this is a distinct effect: under the same data and final output space, deeper chains consistently improve ASR (30.0 → 42.6 → 48.9, Table 2), and more informative constraints yield larger gains (Grounding 42.6 > Motion 39.2). On top of this, multi-turn RL and hierarchical memory—which have no counterpart in OpenHA—make this reasoning adaptive (Figure 5; ASR 48.9 → 62.3) and scalable (context reduced 75%, Table 3).
>
> > W2: *"Method heavily dependent on hand-designed hierarchy beyond Minecraft."*
>
> We share this concern. To provide direct evidence, we have applied DeepHA to Language Table (Lynch et al., 2023), a continuous 2-DoF control task unrelated to Minecraft. Starting from the raw (x, y) displacement space, we constructed Motion and Grounding layers via the same temporal-alignment principle described in the paper, and trained all models with under 1B tokens.
>
> | Task | VLA | HA(Motion) | HA(Grounding) | DeepHA |
> |:-:|:-:|:-:|:-:|:-:|
> | BlockToBlock | 83±9 | 76±8 | 82±7 | **92±4** |
> | BlockToRelLoc | 82±5 | 83±7 | 85±5 | **93±5** |
> | SeparateBlocks | 96±2 | 98±1 | 94±2 | **98±2** |
>
> The results above show that DeepHA consistently outperforms both flat VLA and single-action-space HA baselines across all three task families (e.g., +9 over VLA on BlockToBlock, +11 on BlockToRelLoc). This suggests that the core idea—composing action spaces as progressive constraint layers—is not Minecraft-specific. That said, we acknowledge that choosing which action-space types to define still requires domain knowledge, a limitation shared by existing embodied-AI systems (RT-1, RT-H). Automating this step is an important direction we plan to explore.
>
> > W3: *"Paper would benefit from clearer presentation of implementation, hierarchy design, and data curation."*
>
> We appreciate this suggestion and agree that the current presentation can be clearer. In the revision, we will expand the appendix to provide a more complete description of the implementation details, the action hierarchy design, and the data curation pipeline. In particular, we will add step-by-step details on how each action space is derived, how hierarchy-structured data is constructed, and how the mixture ratios and routing policies are implemented in practice. We will also release the data-processing, training, and evaluation code, together with the curated datasets or anonymized construction scripts/checkpoints where necessary, to improve transparency and reproducibility.
>
> > Q1: *"Provide a direct comparison listing differences relative to OpenHA?"*
>
> Yes. We summarize the key differences below:
>
> | Aspect | OpenHA | DeepHA |
> |---|---|---|
> | Core question | *Which* action space to use | *How to progressively prune* a combinatorial action space via structured reasoning |
> | Hierarchy depth | Fixed 2-level | Scalable N-level action pyramid (Table 2) |
> | Long-horizon memory | N/A | KV-cache reuse, 7,948 → 1,976 tokens (Table 3) |
>
> > Q2: *"How is the action hierarchy constructed in practice? How much supervision is required?"*
>
> In practice, the hierarchy is constructed by aligning different action abstractions to the same underlying interaction trajectory. The key principle is temporal-semantic coverage: if one abstract action consistently spans a longer interaction segment and semantically subsumes another action defined on the same trajectory, we place it at a higher level in the hierarchy. For example, a skill may cover a sequence of grounding actions, and each grounding action may in turn cover multiple motion or raw actions. This means that the cross-level parent-child relations are derived automatically from shared trajectory alignment rather than manually annotated one by one. In the current Minecraft setup, we still reuse existing action interfaces and controllers from prior work, but the hierarchy relations themselves can be constructed from aligned trajectories with limited additional supervision beyond the base trajectories and the per-level action definitions.
>
> > Q3: *"How general is this approach beyond Minecraft? How much effort to transfer?"*
>
> As mensioned in W1, the main transfer cost lies in defining the domain's action interfaces, not in changing the framework. The LangTable results show this can be done with moderate effort.

---

> > ### Author Rebuttal · Reviewer_hNyV · 2026-04-04
> >
> > Thank you for the rebuttal. My main concerns about the distinction from OpenHA, the dependence on a hand-designed hierarchy, and the clarity of the implementation/data construction were adequately addressed. In particular, the rebuttal makes the methodological difference to OpenHA clearer, provides additional evidence beyond Minecraft via Language Table, and commits to improving implementation details and release artifacts in the revision. I look forward to seeing these clarifications, especially the clearer comparison to OpenHA and the expanded details on hierarchy construction and data curation, reflected in the revised paper. Based on this response, I consider my concerns resolved and will raise my score from 3 to 4.

---

> > > ### Author Response · Authors · 2026-04-07
> > >
> > > Thank you for the thoughtful review. We are glad that the rebuttal clarified the distinction from OpenHA and that the Language Table experiments helped address the generalizability concern. We will make sure the revised paper reflects all the improvements discussed, including the expanded details on hierarchy construction and data curation, and the release artifacts.
> > >
> > > As a gentle reminder, if you have not yet had the chance to update your score in the system, we would appreciate it when convenient. Thank you again for the constructive feedback.

---

### Official Review · Reviewer_WTQJ · 2026-03-13

**Soundness:** 2
**Presentation:** 2
**Significance:** 3
**Originality:** 2
**Overall Recommendation:** 3
**Confidence:** 3

**Summary:**

This paper proposes the Deep Hierarchical Agent (DeepHA) for solving a suite of Minecraft tasks. The authors mention that prior approaches are based on pre-defined action spaces, which may be sub-optimal. in contrast, they propose a multi-level action hierarchy, where a VLM policy generates actions at various abstractions levels. These generated actions are provided to the lower level policy to generate primitive actions. The paper also propose an action hierarchy reasoning framework, where the agent uses higher-level abstract actions as thoughts to guide low-leval actions generation. The evaluation is done on a large, proprietary benchmark of over 800 Minecraft tasks, where DeepHA gives state of the art performance.

**Compliance With Llm Reviewing Policy:**

Affirmed.

**Final Justification:**

The rebuttal and Language Table experiments are appreciated and partially address generalizability concerns. However, my two primary concerns remain: novelty over OpenHA is still limited, as both works use hierarchical action chaining with VLMs in Minecraft, and the action spaces and hierarchy still require substantial domain-specific engineering for new tasks.

**Key Questions For Authors:**

1. Since the action abstractions are heavily hand-coded here, can the authors comment on what would make their proposed approach generalizable over other scenarios like robotic manipulation tasks.
2. In Table 1, which inference model was used: direct, eager or greedy?

**Limitations:**

Please see weaknesses above.

**Strengths And Weaknesses:**

Strengths:
1. The empirical evaluations are thorough. The authors have done impressive action abstraction generation and data engineering to make the approach work well on the Minecraft tasks.
2. The ablation studies are thorough which clearly depict the advantages of action abstractions.

Weaknesses:

1. The main contribution over prior OpenHA approach is memory-efficient mechanism and that it uses a mixture of heterogeneous action spaces (ranging from semantic skills to coordinates, motion, and raw actions). However, both papers focus on generating actions through a hierarchical structure (using higher-level semantic goals to guide low-level raw actions), so the novelty is limited.
2. The hierarchy is entirely hand-crafted (a fixed, domain-specific engineering choice for Minecraft), with no method for the model to automatically learn the hierarchy on new tasks. The action spaces are also heavily hand-coded.
3. The strong empirical results are due to massive data engineering rather than the core algorithmic contributions. Further, the paper is hard to reproduce due to reliance on proprietary data requirements.
4. The memory-efficient mechanism is only for inference time, and while it still has to face this memory bottleneck issue while training.

---

> ### Author Rebuttal · Authors · 2026-03-31
>
> > W1: *"the novelty is limited."*
>
> We thank the reviewer for this careful comparison. Although both works involve hierarchical agents, the action hierarchy serves a fundamentally different purpose in DeepHA. While prior work primarily studies which action space to use for a given task, DeepHA addresses a different question: how to tame the combinatorial explosion of low-level action spaces through structured, progressive reasoning.
>
> The raw action space in Minecraft exceeds 16 million possibilities, making direct prediction extremely challenging. DeepHA generates a multi-level action chain (Skill → Grounding → Motion → Raw) where each level provides a structured constraint that prunes the search space for the next. This progressively converts a massive, flat space into a sequence of tractable, low-entropy decisions. Our ablations isolate this effect: with the final output fixed to Raw Action, each additional constraint layer improves ASR—from 30.0 (depth 1) to 39.2 / 42.6 (depth 2) to 48.9 (depth 3)—and more informative constraints yield larger gains (Grounding 42.6 > Motion 39.2), confirming that the improvement stems from constraint quality rather than computation alone (Table 2).
>
> Beyond SFT, multi-turn RL enables the agent to autonomously discover *how deep* to reason and *where to stop*, transforming the hierarchy from a fixed template into an adaptive reasoning process. RL training leads the model to spontaneously generate deeper chains (Figure 5), yielding a substantial further gain (Mine Blocks ASR: 48.9 → 62.3, Table 1). To keep this deep reasoning practical in long-horizon tasks, our hierarchical memory mechanism compresses completed low-level history while preserving active high-level goals via KV-cache reuse, reducing average context from 7,948 to 1,976 tokens without performance loss (Table 3). Together, these contributions move beyond action-space selection toward an inference-time scaling paradigm over action hierarchies.
>
> > W2: *"The hierarchy is entirely hand-crafted... The action spaces are also heavily hand-coded."*
>
> We agree that the Minecraft instantiation uses manually designed action spaces—a deliberate choice to reuse established controllers and isolate the effect of our reasoning and memory mechanisms. That said, the DeepHA framework itself does not require a fixed hierarchy. The cross-level parent-child relations are derived automatically via temporal alignment over shared trajectories (Section 2.2, Appendix B.3), and the framework can also discover hierarchy bottom-up via LLM-driven heuristics. The key requirement is only that multiple typed action spaces exist and can be temporally aligned.
>
> To provide direct evidence, we have conducted new experiments on Language Table (Lynch et al., 2023), a continuous 2-DoF control task. All models were trained on under 1B tokens:
>
> | Task | VLA | HA(Motion) | HA(Grounding) | DeepHA |
> |:-:|:-:|:-:|:-:|:-:|
> | BlockToBlock | 83±9 | 76±8 | 82±7 | **92±4** |
> | BlockToRelLoc | 82±5 | 83±7 | 85±5 | **93±5** |
> | SeparateBlocks | 96±2 | 98±1 | 94±2 | **98±2** |
>
> DeepHA achieves consistent improvement across all tasks (+11 on BlockToRelLoc), demonstrating that the progressive-pruning paradigm transfers beyond Minecraft once task-appropriate action interfaces are provided.
>
> > W3: *"Strong empirical results are due to massive data engineering... hard to reproduce."*
>
> The depth ablation in Table 2 directly controls for data: all depth variants are trained under the same setting, with the only variable being the action chain structure. The ASR improvement from 30.0 to 48.9 therefore cannot be attributed to data engineering. Similarly, Eager mode consistently outperforms Direct mode under the same outcome action space (Table 4), further confirming that the gain comes from the reasoning structure itself. On reproducibility, we will open-source the data-processing pipeline, training and evaluation code, curated datasets, and model checkpoints.
>
> > W4: *"The memory-efficient mechanism is only for inference time; training still faces the bottleneck."*
>
> The same principle is applied during training via causal attention masks (Section 3.3). When a high-level action persists across multiple low-level steps, we mask out completed sub-step history so the model only attends to the active high-level goal and the current observation. This shortens the effective attention span per training sample and ensures training-inference alignment—the model learns under the same compressed-context regime it operates in during deployment.
>
> > Q1: *"generalizability to other scenarios?"*
>
> As discussed in W2, our LangTable results confirm that the framework transfers with minimal adaptation.
>
> > Q2: *"Which inference model was used in Table 1?"*
>
> The DeepHA results in Table 1 are obtained with Greedy mode, i.e., full hierarchical reasoning down to raw actions. Direct and Eager variants are reported separately in Table 4.

---

> > ### Author Rebuttal · Reviewer_WTQJ · 2026-04-04
> >
> > I thank the reviewer for the detailed response, and I appreciate the new Language Table experiments, which implements the approach beyond Minecraft and shows good gains. However I still have two primary concerns: 1. Sufficient novelty over OpenHA, and 2. hand-designed actions spaces and hierarchy, which would require substantial effort to implement on new tasks. I will keep track of other reviewer's scores throughout the rebuttal phase and update my score accordingly.

---

> > > ### Author Response · Authors · 2026-04-07
> > >
> > > Thank you for acknowledging our rebuttal and for the positive comments on the Language Table experiments. We would like to briefly respond to the two remaining concerns.
> > >
> > > Regarding novelty, as discussed in our earlier response, the two works ask different questions: OpenHA studies *which* action space to use, while DeepHA addresses *how to tame the combinatorial explosion*. We hope the depth ablations and RL scaling results in our rebuttal help illustrate that DeepHA's progressive pruning serves a different purpose from OpenHA's action-space selection. We will work to make this distinction clearer in the revision.
> > >
> > > For the hand-designed concern, the action spaces we use (skills, grounding, motion) are not our own designs but standard abstractions that have long been used in the embodied AI community. The hierarchy itself is constructed automatically via temporal alignment, not manually annotated. The Language Table results also show that adapting to a new domain requires moderate effort, not a full redesign.
> > >
> > > We appreciate your feedback throughout this process and will incorporate it to improve the paper.

---

### Decision · Program_Chairs · 2026-04-30

**Decision:**

Accept (regular)

**Comment:**

The authors introduce the Deep Hierarchical Agent (DeepHA), a Vision-Language-Action (VLA) framework that addresses the planner-actor disconnect by generating progressive, multi-level action chains (from high-level skills down to raw motor commands). The paper also introduces a novel hierarchical memory mechanism that dynamically compresses historical context via KV-cache reuse, reducing context length by ~75% for long-horizon tasks. Evaluated on a massive suite of over 800 Minecraft tasks, DeepHA achieves state-of-the-art performance.

## Synthesis of Reviews & Justification for Acceptance

The reviewer consensus was split, with scores leaning borderline. Reviewers praising the work highlighted the impressive systems engineering, the intuitive action-chain reasoning framework, and the massive scale of the empirical evaluation. Reviewers leaning toward rejection raised valid concerns regarding algorithmic novelty (specifically similarities to OpenHA), the reliance on hand-crafted domain-specific action spaces, and the prohibitive compute barrier for reproducibility (over 4,000 A800 GPU-hours).

Despite the borderline scores, I would recommend Acceptance for the following reasons:

### Progressive pruning vs. action selection

The authors' rebuttal successfully clarified the core algorithmic distinction from prior work like OpenHA. DeepHA does not merely select an action space; it uses deep action hierarchies to progressively prune a combinatorial action space (over 16 million possibilities). The depth ablations clearly prove that deeper reasoning chains directly correlate with higher success rates.

### Systems and memory innovation

The hierarchical memory mechanism—which preserves active high-level goals while compressing redundant low-level history—is a highly practical and significant contribution to solving context-window bloat in long-horizon agentic tasks.

### Out-of-domain generalization

The authors mitigated concerns about Minecraft-specific overfitting by providing new, convincing experiments on the 2-DoF Language Table robotic task, demonstrating that the progressive-pruning paradigm transfers effectively to other continuous control domains.

### Community value

While the compute required for Stage 1 pre-training is vast, the authors have committed to open-sourcing their data-processing pipeline, curated datasets, model checkpoints, and a scaled-down training recipe.


## Conclusion

While the algorithmic novelty might be viewed as a synthesis of existing techniques (CoT, MoE, hierarchical RL), the integration is exceptionally well-executed. The sheer scale of the evaluation, combined with the practical innovations in context management, makes this a valuable and highly impactful contribution to the embodied AI community.